

# Attributing the 2017 Bangladesh floods from meteorological and hydrological perspectives

Sjoukje Philip[1], Sarah Sparrow[2], Sarah F. Kew[1], Karin van der Wiel[1], Niko Wanders[3,4], Roop Singh[5], Ahmadul Hassan[5], Khaled Mohammed[2], Hammad Javid[2, 6], Karsten Haustein[6], Friederike E. L. Otto[6], Feyera Hirpa[7], Ruksana H. Rimi[6], AKM Saiful Islam[8], David C. H. Wallom[2], and Geert Jan van Oldenborgh[1]

[1]Royal Netherlands Meteorological Institute (KNMI), De Bilt, The Netherlands
[2]Oxford e-Research Centre, Department of Engineering Science, University of Oxford, United Kingdom
[3]Department of Physical Geography, Utrecht University, Utrecht, The Netherlands
[4]Department of Civil and Environmental engineering, Princeton University, Princeton, NJ, U.S.A.
[5]Red Cross Red Crescent Climate Centre, The Hague, the Netherlands
[6]Environmental Change Institute, Oxford University Centre for the Environment, Oxford, United Kingdom
[7]School of Geography and the Environment, University of Oxford, United Kingdom
[8]Bangladesh University of Engineering and Technology, Dhaka, Bangladesh

**Correspondence:** Sjoukje Philip (philip@knmi.nl) and Geert Jan van Oldenborgh (oldenbor@knmi.nl)

**Abstract.** In August 2017 Bangladesh faced one of its worst river flooding events in recent history. This paper presents for the first time an attribution of this precipitation-induced flooding from a combined meteorological and hydrological perspective. Experiments were conducted with three observational data sets and two climate models to estimate changes in extreme 10-day precipitation event frequency over the Brahmaputra basin. The precipitation fields were then used as meteorological input for

four different hydrological models to estimate the corresponding changes in river discharge, allowing for comparison between approaches and for the robustness of the attribution results to be assessed.

In all three observational precipitation data sets the climate change trends for extreme precipitation similar to observed in August 2017 are not significant, however in two out of three series, the sign of this insignificant trend is positive. One climate model shows a significant positive influence of anthropogenic climate change, whereas the other simulates a cancellation

between the increase due to greenhouse gases and a decrease due to sulphate aerosols. Considering discharge rather than precipitation, the hydrological models show that attribution of the change in discharge towards higher values is somewhat less uncertain than for precipitation, but the 95% confidence interval still encompasses no change in risk. For the future, all models project an increase in probability of extreme events at 2° C global heating since pre-industrial times, becoming more than 1.7 times more likely for high 10-day precipitation, and about a factor 1.5 more likely for discharge. Our best estimate on the

trend in flooding events similar to the Brahmaputra event of August 2017 is derived by synthesizing the observational and model results: We find the change in risk to be greater than one and of similar order of magnitude (between 1 and 2) for both the meteorological and hydrological approach. This study shows that, for precipitation-induced flooding events, investigating changes in precipitation is useful, either as an alternative when hydrological models are not available, or as an additional measure to confirm qualitative conclusions. Besides, it highlights the importance of using multiple models in attribution studies,





particularly where the climate change signal is not strong relative to natural variability or is confounded by other factors such as aerosols.

## 1 Introduction

In August 2017 Bangladesh faced one of the worst river flooding events in recent history, with record high water levels, and the Ministry of Disaster Management and Relief reporting that the floods were the worst in at least forty years. Due to heavy local rainfall, as well as water flow from the upstream hills in India, the water levels in the various rivers in northern Bangladesh burst their banks. This led to the inundation of river basin areas in the northern parts of Bangladesh, starting on 12 August and affecting over 30 districts. The National Disaster Response Coordination Centre (NDRCC) reported around 6.9 million people affected, with 114 people reported dead and at least 297,250 people displaced. Approximately 593,250 houses were destroyed, leaving families displaced in temporary shelters.

Bangladesh is a highly flood-prone country, with flat topography and many rivers that regularly flood and are used to irrigate crops and for fishing. The August 2017 floods were particularly impactful as they followed two earlier flooding episodes in late March and July that year, increasing people's vulnerability. Nearly 85% of the rural population in Bangladesh works directly or indirectly with agriculture and rice is the main staple food contributing 95% to total food production. As is typical after such a flooding, farmers started to plant *aman*, the monsoon rice that is almost entirely rain-dependent. However, the August flood was worse than that of July, and areas such as Dinajpur and Rangpur that normally do not flood were also flooded, see Fig. 1. These are areas that contain significant rice production. As a result, 650,000 hectares of cropland were severely damaged during the August monsoon flooding in the year. *Aman* rice is historically the most variable and yields tend to drop dramatically during major flood years (Yu et al., 2010). The flood-induced crop losses in 2017 resulted in the record price of rice, negatively affecting livelihood and food security. Beyond impacts to agriculture, the floods destroyed transport infrastructure such as railways lines, bridges, and roads, leaving some areas inaccessible for disaster relief efforts. The rise in water and strong current breached roads and embankments and swept away livestock, houses, and assets that may have otherwise been protected. At least 2,292 schools were damaged affecting education for weeks, and 13,035 cases of waterborne illnesses were reported in the aftermath of the floods.

The 2017 flood was markedly different from previous major flood events in 1988 and 1998, when both the Ganges and Brahmaputra flooded simultaneously (Webster et al., 2010). Based on forecasts it was feared that a similar event would occur in 2017, but in this case, swelling of the Brahmaputra, its tributary the Atrai, and the Meghna caused flooding. The worst impacts were along the main reach of the Brahmaputra river (Fig. 1b).

The first estimates of the return period provided by the Bangladesh Water Development Board (BWDB) for the 2017 flood event range from a once in 30 year event to a once in 100 year event, depending on the data source: water level and discharge data at Bahadurabad (the main station for discharge representing the Brahmaputra in Bangladesh) and the flooding forecast system GloFAS. These estimates however, were implicitly based on the assumption of a stationary climate and did not account for the possibility that the frequency of such flooding events may be changing.





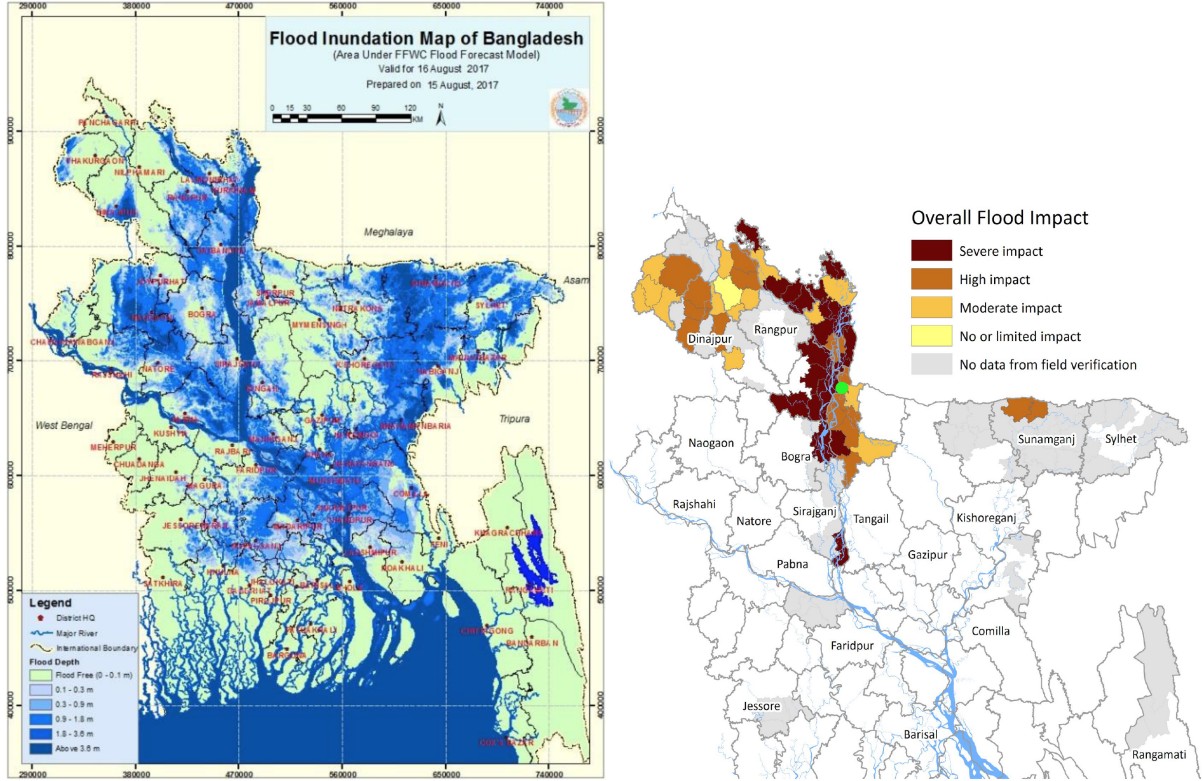

**Figure 1.** Left: inundation forecast map of Bangladesh for 16 August 2017. Right: Overall flood impact of the August 2017 flooding as stated on August 21. The green circle denotes the location of Bahadurabad. The Brahmaputra basin is outlined in Fig. 3. See the original documents[1] for more details on the maps and legends.

Extreme rainfall events that subsequently lead to widespread flooding, such as the 2017 event in Bangladesh, are one of the main types of extreme weather events we are expecting to see more of in a warming climate. But with rainfall not only driven by thermodynamic processes but also affected by changing atmospheric processes it is not a priori clear if such events at a particular location will increase in likelihood or if the dynamic changes will mean that the overall chance of extreme

5 rainfall decreases there (Otto et al., 2016). Furthermore, in the current climate, drivers other than greenhouse gases often play a role that is currently difficult to quantify but likely to mask or exacerbate the effect of greenhouse gas emissions so far on the occurrence likelihood of extreme rainfall events (e.g., aerosols, van Oldenborgh et al., 2016). Hence regional attribution studies are necessary to identify whether and to what extent extreme rainfall events are changing and to provide insight into which drivers have been contributing to those changes and if the trend is likely to continue into the future. Attribution studies require

10 both observational data and models to fully estimate the impact of changes in the climate system. The reported advances in

---

[1]https://reliefweb.int/sites/reliefweb.int/files/resources/SitRep_2_Bangladesh%20Flood_16%20August%202017.pdf and https://reliefweb.int/sites/reliefweb.int/files/resources/72%20hrs-Bangladesh_Flood_Version1_Final%2008212017.pdf





model development for the Brahmaputra region and their success in forecasting gives good confidence in the models' ability to accurately represent the region.

Hydrological models are increasingly used for studies on flooding in Bangladesh. As upstream flow data is absent for Bangladesh, a lot of effort has been made to develop flood forecasting systems based on satellite data and weather predictions. Webster et al. (2010) for instance developed a system that forecasts the Ganges and Brahmaputra discharge into Bangladesh in real time on 1- to 10-day time horizons. In a recent study Priya et al. (2017) show that, using a new long lead flood forecasting scheme for the Ganges-Brahmaputra-Meghna basin, skillful forecasts are provided that inherently express not only a prediction of future water levels but also supply information on the levels of confidence with each forecast. Hirpa et al. (2016) used reforecasts to improve the flood detection skill of forecasts.

Previous scientific studies generally show an increasing trend in climate projections of extreme rainfall and high discharge in the region. For example, Gain et al. (2011) use the PCR-GLOBWB model with input from 12 global circulation models (GCMs, 1961-2100) from the CMIP3 ensemble (Meehl et al., 2007) in a weighted ensemble analysis. They show that in this ensemble, there is a positive trend in peak flow at Bahadurabad: in this model configuration and under the SRES B2 scenario, a peak flow that currently occurs every 10 years will occur at least once every two years during the time period 2080–2099. Dastagir (2015) gives an overview of the change in flooding according to the IPCC 5th Assessment Report and using 16 GCMs from the CMIP5 ensemble (Taylor et al., 2011). They state that the warmer and wetter climate predicted for the Ganges-Brahmaputra-Meghna basins by most climate-related research in this region indicates that vulnerability to severe monsoon floods will increase with climate change in the flood prone areas of Bangladesh. The same conclusion is reached by CEGIS (2013), who use GCM projections and a hydrological model to show that in the wet season an increase in precipitation and annual flow is projected. In line with this, Mohammed et al. (2017) find that in a 2.0 °C warmer world floods will be both more frequent and of greater magnitude than in a 1.5 °C warmer world in Bangladesh, using the hydrological model SWAT with input from the CORDEX regional model ensemble. Zaman et al. (2017) use two sets of climate models with RCP8.5 climate change runs as input in a basin model that simulates flows in major rivers of Bangladesh, including the Brahmaputra. Using the two climate model runs as input they find agreement in the basin model runs for Brahmaputra flow in a 2.0 °C warmer world – one run shows a slightly higher impact of climate change compared to the other run, with an overall increase in monsoon flow of approximately 15% and 10% in the dry season.

Attribution studies on flooding, using both observational data and models, have often been done with precipitation only. In such studies, (e.g., Schaller et al., 2014; van der Wiel et al., 2017; Philip et al., 2018; van Oldenborgh et al., 2017; Risser and Wehner, 2017) it is assumed that precipitation is the main cause of the flooding. For shorter time scales and the relatively small basins involved, this is a reasonable assumption. The major basins in Bangladesh, however, are substantially larger and have longer water travel times than the basins considered in the above studies. Therefore using precipitation alone as a proxy for flooding might not be appropriate. In this paper we explicitly test this assumption by performing an attribution of both precipitation and discharge as a flooding-related measure to climate change. We use observational precipitation and discharge data and a combination of GCMs and hydrological models. To compare the differences between both attribution methods



we calculate the return periods and risk ratios for the August 2017 flooding event in Bangladesh for both precipitation and discharge in observations and models, for past, present and future.

Bangladesh is influenced by three large river basins: the Ganges basin in the northwest, the Brahmaputra basin in the northeast and the Meghna basin in the east. During the monsoon season the rainfall moves northwest across the country, starting

in May-June-July in the Meghna basin. Usually two to three weeks after peak rainfall in July, the rivers in the Brahmaputra basin reach their peak discharge. Finally, in August and September the Ganges basin river discharge peaks. The largest impact of flooding in August 2017 was felt in the northern parts of Bangladesh (Fig. 1). As this was mainly caused by precipitation in the Brahmaputra basin, the focus in this paper will be on this basin. In the Brahmaputra basin little water originates from precipitation on the northern side of the Himalaya (China/Tibet), with most of the water coming from precipitation in the

upstream Assam region in India. Precipitation in Bhutan also contributes to the river water in Bangladesh.

In this paper we use two event definitions: one based on precipitation and one based on discharge. Both observational data and model data can be used for these two event definitions. For precipitation we average over the whole Brahmaputra basin and take a 10-day average, as the largest precipitation volume in the Brahmaputra basin travels to Bangladesh within 10 days. Only precipitation in July-August-September (JAS) is analysed as it is only in these months that precipitation is considered the

major cause of flooding. For discharge we simply use the daily maximum discharge at Bahadurabad, a station situated to the north of the confluence point of the Ganges with the Brahmaputra, in JAS.

The data and methods used are described in Section 2. Sections 3 and 4 describe the analysis for observations and models respectively. The results are synthesized in Section 5. A discussion follows in Section 6 and the paper ends with some conclusions.

## 20  2  Data and methods

### 2.1  Observational data

The first observational dataset we use is the 0.5° gauge-based CPC analysis 1979-now (www.cpc.ncep.noaa.gov/products/ Global_Monsoons/gl_obs.shtml). This is the longest gauge-based daily gridded dataset available that is still being updated. The seasonal cycle of precipitation in the Brahmaputra basin is shown in Fig. 2a. Monsoon rains start rising slowly with a

maximum in July and August, and become less from September onwards. As precipitation will not, in general, cause flooding before July, we will use the months JAS for the precipitation analysis.

The second gauge-based dataset we use for comparison is the 1.0° GPCC dataset (1988-now) (ftp://ftp-anon.dwd.de/pub/ data/gpcc/html/download_gate.html). As this is a much shorter dataset we expect the signal to noise ratio in the trend to be smaller. We only use this dataset to additionally check the observations. The seasonal cycle can be found in Fig. S1.

The third dataset is the reanalysis dataset ERA-interim (ERA-int, 1979-now) (http://www.ecmwf.int/en/research/climate-reanalysis). Precipitation of this dataset is analysed directly. As well as precipitation, temperature and potential evapotranspiration (calculated, Penman-Monteith) are used to drive one of the hydrological models see Section 2.2.2. The seasonal cycle of ERA-int can be found in Fig. S1.





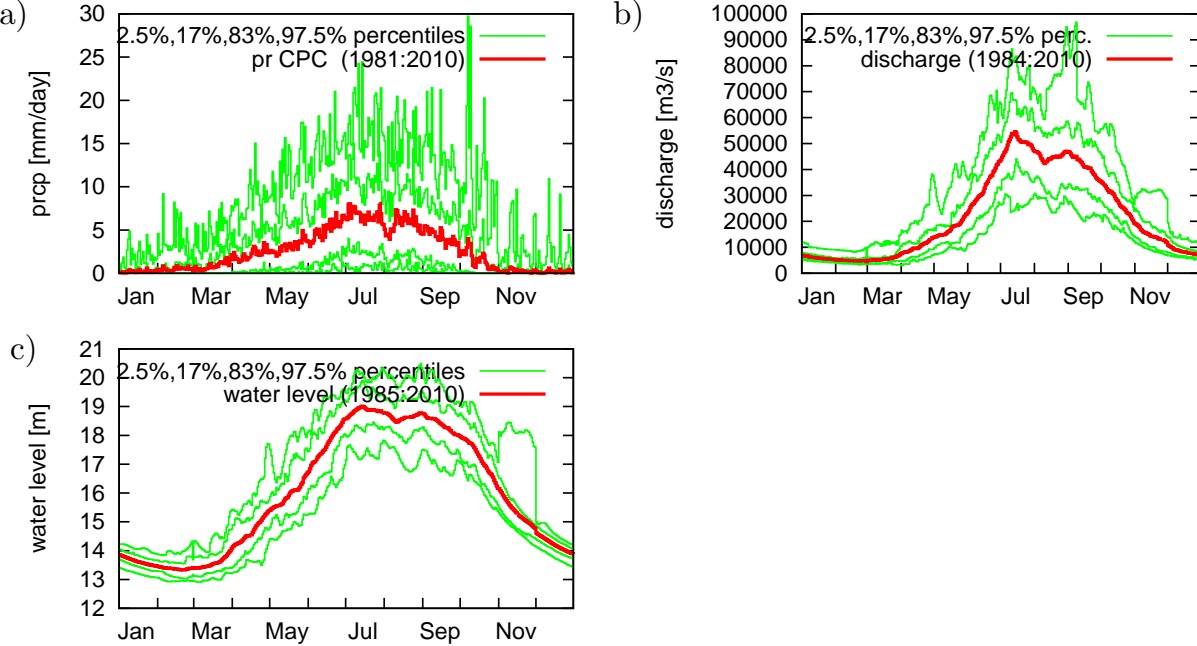

**Figure 2.** Seasonal cycle of (a) precipitation in the Brahmaputra basin for CPC, (b) discharge at Bahadurabad and (c) water level at Bahadurabad. The red line shows the mean value, green lines show the 2.5, 17, 83 and 97.5 percentiles.

We use discharge and water level data from Bahadurabad. Discharge data are available for the years 1984-2017 and water level for the years 1985-2017 (source: BWDB). For both datasets the seasonal cycle is shown in Fig. 2b,c. Additionally, we have a discharge dataset for the years 1956-2006 (source: BWDB). As the rating between water level, velocity and discharge is not exactly the same in the two discharge datasets, we consider a simple merge of the datasets not to be appropriate. The
5   1984-2017 dataset is used in the analyses, but results are compared to calculations with the 1956-2006 dataset and merged datasets.

## 2.2 Model descriptions

Both the global circulation model and regional model that are used for the analysis of precipitation are described in Section 2.2.1. Details on the hydrological models can be found in Section 2.2.2 with further details on the hydrological models,
10  including validation and calibration, described in the Supplement.

### 2.2.1 Precipitation

First, we performed the same analysis as in the observations with ensemble experiments from the coupled atmosphere-ocean general circulation model EC-Earth 2.3 (Hazeleger et al., 2012). The resolution of the model is T159, which is about 125 km.





We use three different experiments. The first one is a transient model experiment, consisting of 16 ensemble members covering 1861–2100 (here we use up to 2017), which are based on the historical CMIP5 protocol until 2005 and the RCP8.5 scenario (Taylor et al., 2012) from 2006 onwards. To compare these runs with observations, we use model years in which the difference in smoothed observed global mean surface temperature (GMST) between 2017 and a year in the past (1984, 1979, and 1900) agrees with the difference in ensemble averaged model-GMST between 2017 and that year in the past. These are the model years 1985 and 1979 (corresponding to 1984 and 1979 in observations) and 1934 (corresponding to 1900 in observations).

The other two experiments are two time slice experiments, based on the above 16-member transient model experiment. Two experimental periods are selected in which the model-GMST is as observed in 2011-2015 ('present-day' experiment, model years 2035-2039) and as pre-industrial (1851-1899) + 2 °C warming ('2°C-warming' experiment, model years 2062-2066). For each time slice, 25 members are generated from each of the 16 transient ensemble members and these are integrated for 5 years, resulting in $16 \times 25 \times 5 = 2000$ years of data for each time slice. The difference in model-GMST between the two time slice experiments is such that it is the same as the difference between the observed present-day GMST and a 2 °C warmer world.

Second, large ensembles of climate model simulations are created using the distributed computing weather@home modelling framework (Guillod et al., 2017; Massey et al., 2014). The weather@home setup consists of the Met Office Hadley Centre Atmosphere-only model, HadAM3P running globally at a resolution of $1.25° \times 1.875°$ to drive the Met Office Hadley Centre Regional Model, HadRM3P running at a resolution of 50 km over South Asia. The model is driven with prescribed sea surface temperatures (SSTs) and sea ice (SIC). Table 1 describes the experiments used in this study, which are grouped into three sets - (i) ensembles for the historical period 1986-2015 (white), (ii) ensembles for 2017 (light gray), (iii) ensembles for assessing possible changes in the future (dark gray).

The first set of experiments captures the years 1986-2015. To derive the value of the observed threshold within the weather@home model a climatology (with each year run independently) from 1986-2015 is run using observed OSTIA SSTs and SIC (Donlon et al., 2012) and CMIP5 historical+RCP4.5 estimates of other forcings (hereafter, Historical). A second climatology representing pre-industrial conditions is run from 1986-2015 using observed OSTIA SSTs and SIC naturalised as above with CMIP5 estimates and pre-industrial CMIP5 forcing conditions (hereafter, Natural). The Natural ensemble is constructed following Schaller et al. (2016), where the anthropogenic signal in the SST is derived from the difference between historical and historicalNat CMIP5 simulations from 13 different models and removed from the observed SSTs. A third ensemble (hereafter, GHG-only) is also included for the years 1986-2015, where GHG emissions follow the same protocol as for the Historical ensemble, but all other forcing components are kept at pre-industrial levels as for the Natural ensemble. As with the Natural component, the signature of GHG emissions in the SST is removed by comparing the relevant CMIP5 simulations to obtain a set of plausible SST patterns. The SSTs are specifically reconstructed to respond to changes in GHG emissions only, without the influence of the historical changes in other anthropogenic forcings.

The second set of experiments simulates the year 2017 for three different ensembles. One ensemble simulates the world as observed for 2017 (hereafter Actual 2017) taking observed OSTIA SST and SIC to drive the model and with sulphate





**Table 1.** Experiments with the weather@home ensemble, including ensemble size and short description.

| Experiment | Ensemble size | Description |
| --- | --- | --- |
| Historical | 5222 | 1986-2015 SSTs and sea ice as observed, other forcings from CMIP5 historical+RCP4.5 |
| Natural | 6659 | 1986-2015, SSTs reconstructed for pre-industrial, all other forcings pre-industrial |
| GHG-only | 4931 | 1986-2015, SSTs reconstructed for GHG emissions only, CMIP5 historical+RCP4.5 GHG emissions, all other forcings preindustrial |
| Actual 2017 | 2996 | 2017 SSTs and sea ice as observed, other forcings as RCP4.5 |
| Natural 2017 | 6126 | 2017, SSTs reconstructed for pre-industrial, all other forcing pre-industrial |
| GHG-only 2017 | 5386 | 2017, SSTs reconstructed for GHG emissions only, RCP4.5 GHG emissions, all other forcings preindustrial |
| Current | 2781 | 2004-2016, SSTs and sea ice as observed, all other forcings from CMIP5 RCP4.5 as per HAPPI experiment design |
| 1.5 Degree | 1848 | Representative decade with 1.5° C of additional warming as per HAPPI experiment design |
| 2.0 Degree | 1892 | Representative decade with 2° C of additional warming as per HAPPI experiment design |

emissions, well-mixed greenhouse gas, solar variability and volcanic emissions taken from CMIP5 values. A counterfactual ensemble (hereafter Natural 2017) is run for 2017 and uses pre-industrial forcing estimates as Natural above. A third ensemble for GHG only in 2017 also generated as above (hereafter GHG-only 2017). The difference between the GHG-only 2017 simulations and the Actual 2017 experiments is the inclusion of the anthropogenic sulphate emissions, so by comparing the two simulations it is possible to derive an estimate of the anthropogenic aerosol influence. We have successfully tested the assumption of linearly additive forcing responses using a small subset of CMIP5 Aerosol only experiments.

A third set of experiments are performed with weather@home 'South Asia' following the HAPPI experimental design (Mitchell et al., 2017) where a decade of simulations are performed representing a world as currently observed (hereafter Current), and with global mean surface temperature increase limited to 1.5° C and 2° C above pre-industrial levels (hereafter 1.5 Degree and 2 Degree respectively). Although the simulation period is only 10 years (adding future SST pattern onto OSTIA SSTs for 2006-2015), the HAPPI model setup is comparable with the climatological and 2017 experiments.

### 2.2.2 Discharge

In this study we used the global hydrological model PCR-GLOBWB 2 (Sutanudjaja et al., 2017). This model was selected because of its ability to simulate the hydrological cycle, including reservoir operations and human water interactions at continental and global scales. The model simulates the global water balance at daily temporal resolution and at either 10 km or 50 km spatial resolution. It resolves the water balance at the surface, by using precipitation, temperature and potential evaporation inputs from meteorological observations or climate models.

We used PCR-GLOBWB to conduct several river discharge simulations. Firstly we checked the performance of the model by comparing the output to observations of (i) the 2017 event, (ii) other historical flooding events and (iii) river discharge



over a historical period, using two observational precipitation datasets as forcing. Secondly, we generated large ensembles of discharge for calculating risk ratio statistics using the EC-Earth climate model experiments as input. The simulations based on observational data were done at 10 km spatial resolution and provide simulated daily discharge for the Brahmaputra river basin. We used CPC and ERA-interim precipitation estimates for the period 1979-2017 to generate daily fields of soil moisture,

groundwater and discharge. The simulations also require temperature and evapotranspiration input, which were taken from ERA-interim for both the CPC and ERA-interim runs. The EC-Earth experiments, i.e. both the 16 transient ensemble members (years 1920-2066 available for 12 members, years 1880-2066 for the other 4 members) and the two time slice experiments ('present-day' and '2°C-warming'), were used as forcing for PCR-GLOBWB at the coarser resolution of 50 km.

Second, we use the Soil and Water Assessment Tool (SWAT), which is a commonly used hydrological model for investigating

climate change impacts on water resources at regional scales (Gassman et al., 2014). This model has already been used to simulate impacts of climate change on the flows of the Brahmaputra River (Mohammed et al., 2017, accepted). The water balance equation used in SWAT consists of daily precipitation, runoff, evapotranspiration, percolation and return flow.

The SWAT model was used in this study to simulate flows by taking inputs from both the transient and time-slice EC-EARTH experiments and weather@home experiments, using daily maximum and minimum temperatures and precipitation.

The parameters of the SWAT model were calibrated twice using climatological data (1986-2015) of each of the two climate models before applying data from the corresponding climate models to simulate flows.

The third hydrological model we use is Lisflood. This is a fully distributed and semi-physically based model initially developed by the Joint Research Centre (JRC) of the European Commission in 1997. It was subsequently updated to forecast floods and analyse impacts of climate and land-use change (Burek et al., 2013). It has been used for operational flood forecasts

as part of the European Flood Awareness System (EFAS) since 2012 (https://www.efas.eu/about-efas.html). Lisflood uses a 1-dimensional channel routing algorithm and solves kinematic wave equations in an implicit manner using four point finite difference solutions to route runoff through the channel network (Knijff et al., 2010). In this study we only used the routing scheme of the model to simulate horizontal water fluxes while the vertical fluxes, surface and subsurface runoff were simulated with the MOSES land surface scheme. The model was run at a spatial resolution of $0.1° \times 0.1°$ and a temporal resolution of a

day. It was calibrated with the Parallel version of the Dynamically Dimensioned Search Algorithm (PDDS) (Tolson and Shoemaker, 2007) using an auto-calibration software, Ostrich (Matott, 2017). The Lisflood model was used in this study to simulate the river flow of the Brahmaputra river at Bahadurabad gauging station with input data from the Weather@home model.

The fourth and final hydrological model used in the analysis is a fully distributed River Flow Model (RFM) that estimates the streamflow by discrete approximation of the one-dimensional kinematic wave equation (Dadson et al., 2011). RFM is designed

to route the gridded runoff simulated by climate models, land surface schemes (MOSES in our case) or rainfall scenarios to generate river flow at daily or hourly temporal resolution. It has a simple fully distributed spatial structure which makes it possible for it to be coupled with other models. RFM routes the overland flow and stream flow in 2 dimensions and can have different wave speeds for surface and subsurface runoffs. RFM was also calibrated with PDDS and Ostrich.





### 2.3 Statistical methods

We use a class-based event definition, i.e., we consider all events that are as extreme or more than the observed event on a one-dimensional scale, in this case 10-day averaged precipitation averaged over the Brahmaputra basin or daily runoff at Bahadurabad.

The first step in an attribution analysis is trend detection: fitting the observations to a non-stationary statistical model to look for a trend outside the range of deviations expected by natural variability. In this case we study the trends of extreme high precipitation and river discharge values. In extreme value analysis, the Generalised Extreme Value (GEV) Distribution (Coles, 2001) is often used to fit and model the tail of the empirical distribution for this type of event, the maximum daily or 10-daily value over the monsoon season. The shape parameter $\xi$ determines the tail behavior; negative indicates light tail while positive

indicates heavy tail behavior. When $\xi = 0$, the distribution simplifies to the Gumbel distribution. Global warming is factored in by allowing the GEV fit to be a function of the (low-pass filtered) global mean surface temperature. In the case of precipitation and discharge extremes, it is assumed that the scale parameter $\sigma$ (the standard deviation) scales with the position parameter $\mu$ (the mean) of the GEV fit. This assumption is also known as the index flood assumption (Hanel et al., 2009) and is commonly applied in hydrology to restrain the number of fit parameters. It can be checked in the model experiments where there is

enough data to fit both $\mu$ and $\sigma$ independently. These parameters are scaled up or down with the GMST using an exponential dependency similar to Clausius-Clapeyron scaling: $\mu = \mu_0 \exp(\alpha T/\mu_0), \sigma = \sigma_0 \exp(\alpha T/\mu_0)$, with $T$ the smoothed global mean temperature and $\alpha$ the trend that is fitted together with $\mu_0$ and $\sigma_0$. The shape parameter $\xi$ is assumed constant. 95% confidence intervals are estimated using a 1000-member non-parametric bootstrap. This approach has been used in several previous attribution studies (e.g. van Oldenborgh et al., 2016; van der Wiel et al., 2017; Otto et al., 2018). This fit also gives

the return periods of the observed event.

The second step in an attribution analysis is the attribution of the detected trend to global warming, natural variability or other factors, such as changes in aerosol concentration or ENSO, and requires comparing model simulations with and without anthropogenic forcing. There are two approaches. The first is to run two ensembles: one with current conditions, and one with conditions as they would have been without anthropogenic emissions. The number of events above the threshold is compared

between the two ensembles. In the second approach, we approximate the counterfactual climate by the climate of the late 19th century and fit the same non-stationary GEV that was described above to the model data. The distribution is evaluated for a GMST in the past and and the current GMST. These two approaches have been used before for studies of extreme precipitation, (e.g., Schaller et al., 2014; van Oldenborgh et al., 2016; van der Wiel et al., 2017; van Oldenborgh et al., 2017).

As a third step, we calculate the risk ratio (RR) or change in probability for different time intervals. These include for

instance the difference between present day and 1979, or between present day and pre-industrial times. For observations we calculate risk ratios with respect to the beginning of the dataset. If possible, we additionally transform these into risk ratios with respect to pre-industrial, in this case set to be the year 1900, such that we can compare this with model runs for pre-industrial settings. For this transformation we assume that the RR depends exponentially on the covariate, in this case the global mean temperature change. For instance if we would find that the probability doubles for 0.5° C warming, we assume that to first





order it would double again for 1° C warming. With future model runs we can also calculate risk ratios between 2° C climate and the climate of now.

A last step in the analysis is the synthesis of the results into a single attribution statement. Though the method for evaluating risk ratios using a transient model or observations is different from that using ensemble time-slice experiments that are explicitly

designed to simulate a +2.0 °C world, we are able to give an average value for all observations and models combined, and assume this gives a good first order estimate of the overall risk ratio.

The differences among the RRs of these ensembles and the observations are due to natural variability, different framings and to model spread. The relative contribution of random natural variability can be estimated from a comparison of the uncertainty derived from each fit with the spread of the different estimates of the RR from observations and models. We do this by

computing a $\chi^2/\mathrm{dof}$, with the number of degrees of freedom, $dof$, one less than the number of fits. If this is roughly equal to one, the variability is compatible with only the natural variability that determines the uncertainty on each separate model estimate of the RR. If it is much larger than one, the systematic differences between the framings and models contribute significantly.

We choose to use a weighted average, with the weights being the inverse uncertainty squared for each RR (models and obser-

vations). The uncertainties are approximated by symmetric errors on $\log(\mathrm{RR})$ and added in quadrature ($\epsilon^2 = \sqrt{\epsilon_1^2 + \epsilon_2^2 + \ldots + \epsilon_N^2}/N$). If there is a significant contribution of $\chi^2$ due to model spread, this has to be propagated to the final result and the final uncertainty is larger than the spread due to natural variability. In this case we choose to give all models equal weight. The method described here was also used in Eden et al. (2016) and Philip et al. (2018).

## 3   Observational analysis

### 3.1   Precipitation

Fig. 3a shows the time series of CPC precipitation averaged over the Brahmaputra basin for 90 days ending at 2 September 2017. The 10-day average at the beginning of July is slightly higher than the 10-day average beginning of August, 14.38 versus 14.20 mm. As we are interested in the August flooding event, we take the precipitation value from the August event, which has a maximum over 5–14 August, see Fig. 3c. The 10-day average annual maximum precipitation is fitted to a GEV distribution.

The return period plots show that the distribution can be described by a GEV by overlaying the data points and fit for the present and a past climate (Fig. 3d). The return period calculated from this fit is 11 years (95% CI (Confidence Interval) 4-200 years) for the current climate. There is a positive trend with a risk ratio with respect to 1979 of a factor 6 ($> 0.3$), although the trend is not significant at $p<0.05$ two-sided (the uncertainty range includes 1).

A similar approach to the one used for CPC data is applied to ERA-int data. In this dataset the July 2017 10-day average

was also just slightly higher than the August 2017 10-day average. The return period for the August event with a value of 17.9 mm/day was 2 years (95% CI 1 to 6 years) in the current climate. This dataset also shows a non-significant positive trend with a risk ratio of 1.9 (0.6 to 7), i.e., a doubling of the probability of an event like this or higher.





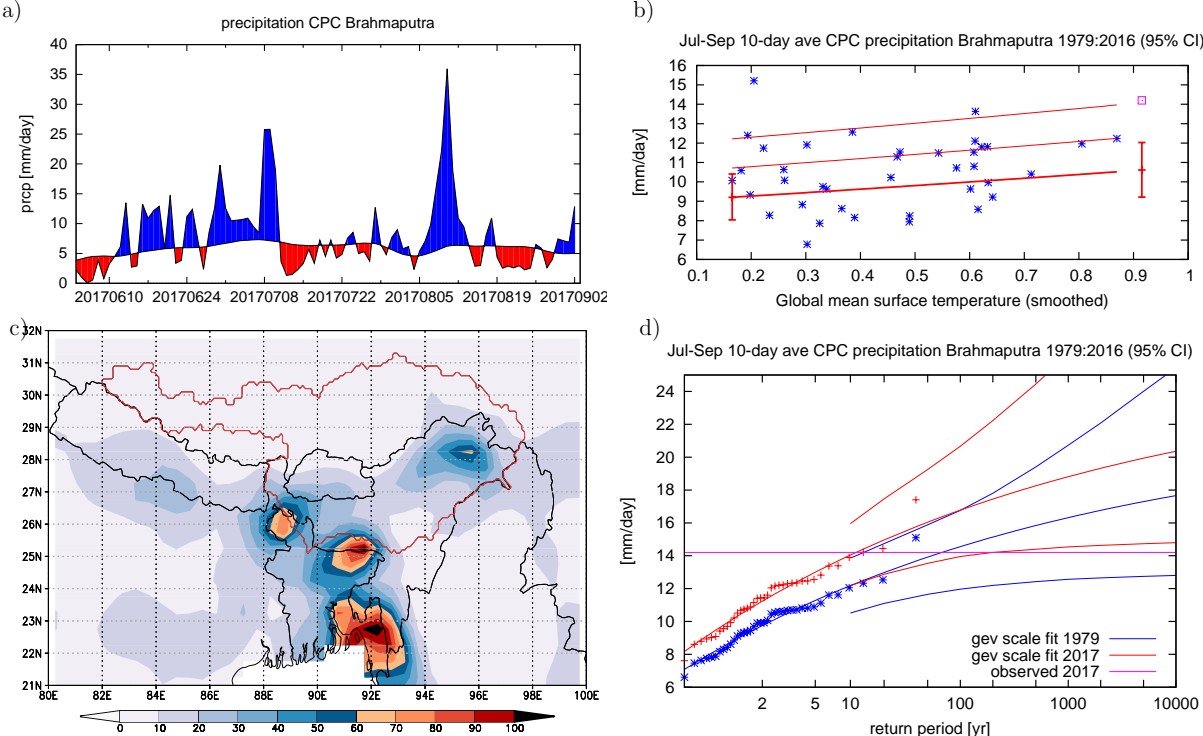

**Figure 3.** CPC data (a and c) and analysis of the highest observed 10-day mean rainfall in the Brahmaputra basin in July–September (b and d). a) time series of precipitation averaged over the Brahmaputra basin: blue is more than average, red less than average. c) 10-day averaged precipitation over the Brahmaputra basin. Dark red means heavy precipitation. In red the contours of the Brahmaputra basin. b) the location parameter $\mu$ (thick line), $\mu+\sigma$ and $\mu+2\sigma$ (thin lines) of the GEV fit of the 10-day averaged data. The vertical bars indicate the 95% confidence interval on the location parameter $\mu$ at the two reference years 2017 and 1950. The purple square denotes the value of 2017 (not included in the fit). d) the GEV fit of the 10-day averaged data in 2017 (red lines) and 1950 (blue lines). The observations are drawn twice, scaled up with the trend (smoothed global mean temperature) to 2017 and scaled down to 1950. The purple line shows the observed value in 2017.





**Table 2.** Return periods and risk ratios for observations of precipitation, discharge and water level. The column RR1 gives results wrt 1979 (precipitation), wrt 1984 (discharge), and 1985 (water level). The column RR (wrt 1900) scales the results to the pre-industrial period.

| variable | dataset (august 2017 value [mm/day]) | RT | 95% CI | RR1 | 95% CI | RR (wrt 1900) | 95% CI |
|---|---|---|---|---|---|---|---|
| precipitation | CPC (14.20) | 11.2 | 4.1 ... 200 | 6.0 | >0.30 | 18 | >0.2 |
| | ERA-int (17.89) | 2.4 | 1.4 ... 6.2 | 1.9 | 0.64 ... 7.2 | 2.8 | 0.5 ... 24 |
| | GPCC 1988-2017 (16.79) | 21 | 4.2 ... 800 | 0.65 | 0.009 ... 30 | 0.5 | 0.004 ... 800 |
| discharge | 1984-2017(78.262) | 4 | 3 ... 6 | 1.3 | 0.1 ? 9 | 0.8 | 0.02 ... 8 |
| water level | 1985-2017 (20.83) | 12 | 3 ... 350 | 170 | >0.6 | | |

Finally, the shorter GPCC dataset gives similar results as well. Risk ratios are given with respect to 1979 in order to compare this with the other datasets. The August 2017 10-day average is slightly higher than the July 10-day average. The return period is about 20 years (95% CI 4-800 years). The risk ratio is not significantly different from one.

The results of return periods and risk ratios based on observations can be found in Table 2. For analyses with models we use the return period from the CPC dataset of 11 years for this event, as based on local experience we think that this is the best estimate. Due to the shape parameter being close to zero the risk ratio will not have a strong dependence on this choice: for a Gumbel distribution it is independent of the return time.

### 3.2 Discharge

The highest discharge in 2017 was reached on the 16th of August, with a value of about 78,000 m3/s. This was clearly higher than any value in July in the same year, as opposed to the precipitation values discussed above. There have been several years in which the discharge was higher than in 2017, including the years 1998 and 1988, which are the two maximum values in the discharge record. The return period is calculated from the discharge dataset since this is our best observational estimate. However it is worth noting that there is a large uncertainty in the accuracy of the discharge measurements from 2012 onwards. We check if the results are robust by comparing the outcomes from the different datasets.

We fitted the discharge time series of Bahadurabad to a GEV distribution. In this distribution we see no trend (95% CI wrt 1900 is 0.1 to 40), see Fig. 4. Therefore we calculate the return period assuming no trend. This results in a return period of the August 2017 event of 4 years (95% CI 3 to 6 years). A cross-check with the 1956–2006 dataset or a merge of the two discharge datasets gives similar results.

### 3.3 Water level

Although we only have water level available in observations and not for models, we still analyze the observational water level time series from Bahadurabad. The highest value in 2017 was on the 16th of August, with a value of 20.83m. This is 1.33m higher than the danger level of 19.50. In contrast to the discharge this was a record level since the beginning of the dataset





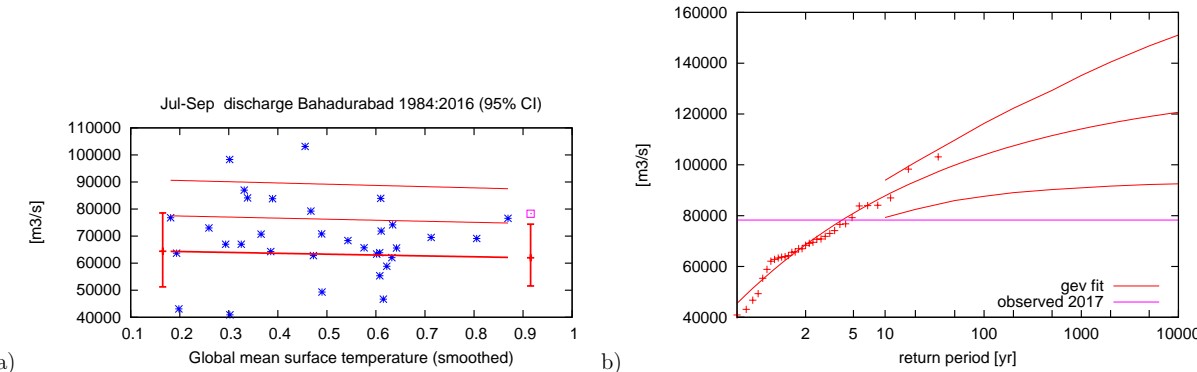

**Figure 4.** Analysis of the highest observed daily discharge at Bahadurabad in July–September. a) the location parameter $\mu$ (thick line), $\mu+\sigma$ and $\mu+2\sigma$ (thin lines) of the GEV fit of the discharge data. The vertical bars indicate the 95% confidence interval on the location parameter $\mu$ at the two reference years 2017 and 1984. The purple square denotes the value of 2017 (not included in the fit). b) the GEV fit of the discharge data assuming no trend. The purple line shows the observed value in 2017.

(1985). It should be noted that the water level is also influenced by factors other than climate change, for instance a raising of the river bed by sedimentation and obstruction of the river channel by man-made constructions.

Under the same assumption as for precipitation and discharge, that water level scales with GMST, the return period in the current climate is estimated to be 12 years (95% CI 3-350 years), see Fig. 5b. However, although the risk ratio between 2017 and 1985 is as large as 170, this is just non-significant with a lower bound of 0.6. This is probably due to the relatively short length of the dataset. In addition, we calculate a return period assuming no trend, see Fig. 5c. This gives a return period of about 80 years (> 25 years, 95% CI). This agrees with the estimates from BWDB.

## 4   Model analysis

### 4.1   Precipitation

In this section we present model validation and analysis results for the precipitation experiments, first for EC-Earth and then for weather@home.

For validation of the EC-Earth 2.3 model we use the years in the transient runs that correspond to the observational years 1979-2017. In the model as expected most precipitation falls in the months JJA, with a peak in July, like in observations, though the increase in precipitation is slightly stronger in June than it is in observations. See Figure S1 in the Supplement. As it is assumed that the scale parameter $\sigma$ scales with the position parameter $\mu$ of the GEV fit, we check whether the dispersion parameter $\sigma/\mu$ and the shape parameter in this model are similar to those calculated from observations. The parameters of the GEV distribution that is fitted from the precipitation of these model years correspond well to the same parameters for CPC data.





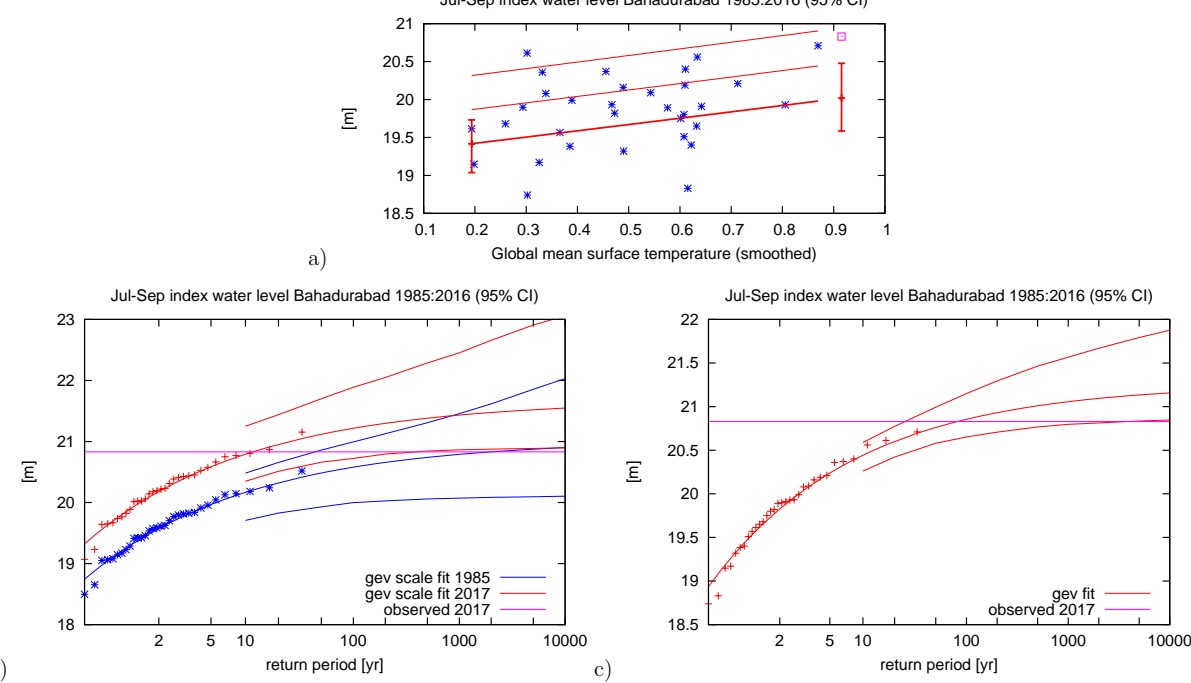

**Figure 5.** Analysis of the highest observed daily water level at Bahadurabad in July–September. a) the location parameter $\mu$ (thick line), $\mu+\sigma$ and $\mu+2\sigma$ (thin lines) of the GEV fit of the discharge data. The vertical bars indicate the 95% confidence interval on the location parameter $\mu$ at the two reference years 2017 and 1985. The purple square denotes the value of 2017 (not included in the fit). b) the GEV fit of the water level data in 2017 (red lines) and 1985 (blue lines), assuming a trend. The observations are drawn twice, scaled up with the trend (smoothed global mean temperature) to 2017 and scaled down to 1985. c) the GEV fit of the same discharge data assuming no trend. The purple line in b) and c) shows the observed value in 2017.

The risk ratio of precipitation is calculated in the same way as for observations, using the data period 1880-2017 such that we can use the same years for the EC-Earth runs and the PCR-GLOBWB and SWAT runs with EC-Earth input, see Fig. 6. The threshold is chosen such that the return period in the current climate is similar to the observed return period when using the same years. The risk ratio between 2017 and pre-industrial is 3.3 (95% CI 2.7 to 4.2) in these transient runs. This corresponds

5   to an increase of intensity for the same return period of 10% (95% CI 9% to 11%). For the future (figures not shown) we calculate return periods from the present and future distributions separately, again following the same statistical method as for observations but with two separate GEV fits that do not depend on GMST. The risk ratio between a 2 °C climate and the present follows from this, with a value of 1.8 (95% CI 1.7 ... 2.1). We thus conclude that in the EC-Earth 2.3 model there is a significant positive trend in the magnitude of precipitation events such as the one in August 2017, both in the past (pre-industrial up to

10   now) and in the future.

For weather@home, we compare the annual cycle of 10-day running mean precipitation (see Fig. S2) and its spatial pattern in the Brahmaputra basin from Historical simulations with CPC and GPCC observational records. As has also been seen



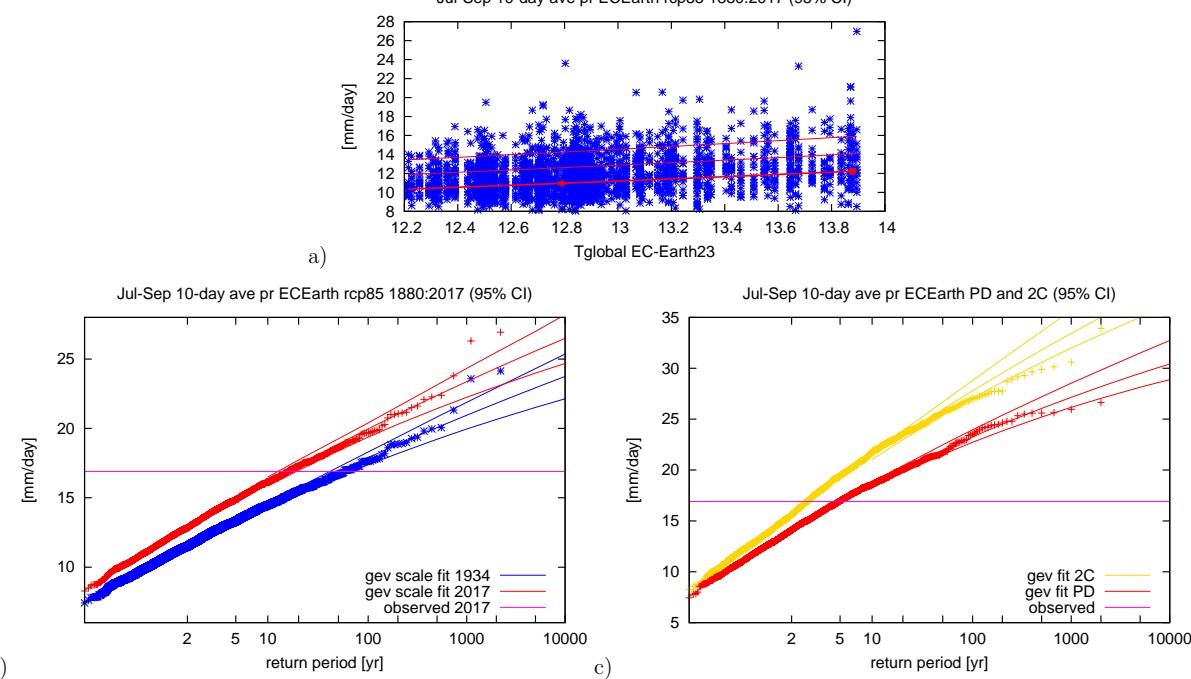

**Figure 6.** Analysis of the highest 10-day average precipitation in July–September in the EC-Earth model over the years 1880-2017. a) the location parameter $\mu$ (thick line), $\mu+\sigma$ and $\mu+2\sigma$ (thin lines) of the GEV fit of the discharge data. The vertical bars indicate the 95% confidence interval on the location parameter $\mu$ at the two reference years 2017 and 1934. b) the GEV fit of the precipitation data in 2017 (red lines) and 1934 (blue lines), assuming a trend. The data is drawn twice, scaled up with the trend (smoothed global mean model temperature) to 2017 and scaled down to 1934. c) GEV fits for the present day (PD, red) and +2°C world (2C, yellow) simulations. The purple lines in b) and c) shows the threshold value for which the risk ratio is calculated.

within other regions of Bangladesh (Rimi et al., 2018a), weather@home rainfall is too intense in the pre-monsoon season but lies within observational uncertainty during the monsoon season itself. Also the variability of 10-day model precipitation is underrepresented by the model for the monsoon season. During the monsoon season the spatial pattern and magnitude of weather@home output agrees well with GPCC and CPC observations (not shown).

5  Fig. 7 shows the return periods of the maximum 10-day precipitation during JAS from the weather@home simulations. The threshold used in this analysis is defined by taking the magnitude from the Historical simulation corresponding to the return period derived from the CPC observational dataset.

Fig. 7a shows the results for the Historical and 2017-specific experiments, which we use to analyse how probabilites may have changed in the period from pre-industrial times up to now. There is no statistically significant difference between the

10  Historical and Natural simulations, with a risk ratio of 0.92 (0.84...1.02).

The difference in return periods between the Historical and Actual 2017 experiments gives an indication of the influence of the natural variability of the SST pattern on the precipitation in this region. The Historical ensemble is driven with 30 years of





differing SST patterns containing different patterns of natural variability such as the El Niño – Southern Oscillation, whereas Actual 2017 uses only the observed 2017 OSTIA SSTs. The SST pattern in 2017 (Actual 2017) made extreme precipitation events less likely than the climatological mean (Historical) with a risk ratio of 0.25 (95% CI 0.2...0.31). Within the set of simulations conditioned on 2017 SSTs, the negligible anthropogenic influence found in the full range SST set is confirmed:

the Actual 2017 and Natural 2017 ensembles also do not show a statistically significant difference and have a risk ratio of 0.97 (95% CI 0.76...1.23), indicating that, if anything, high precipitation events similar to the amplitude observed are more prevalent in our model in the Natural ensemble, whether or not conditioned on 2017 SST conditions.

To understand this result more fully it is useful to look at the 'GHG only' simulations in Fig. 7a (compare GHG-only with Historical, and GHG-only 2017 with Actual 2017). The GHG-only simulations show that increased GHG emissions have

increased the likelihood of this kind of event (relative to the Natural simulations) but that when the sulphate aerosol emissions are taken into account (in the Historical and Actual 2017 simulations), we find a counterbalancing effect that acts to reduce rainfall and hence the risk for severe flooding. This effect has also been noted by van Oldenborgh et al. (2016); Rimi et al. (2018b). Within the weather@home model sulphate emissions are included, although emissions due to other important aerosols such as black carbon, which can counteract sulphate effects, are not represented. The aerosol effect in HadRM3P is therefore

potentially overestimated. The results do highlight the non-linear change in risk over time as a function of anthropogenic aerosol emissions. EC-Earth follows the historical+RCP8.5 protocol for aerosols, and includes both sulphate emissions, black and organic carbon. It does not include any indirect aerosol effects. The differences in aerosol representation and model handling of aerosols, as well as the influence of the experimental configuration on aerosol concentration, between EC-Earth and weather@home may account for the difference in risk ratios for the past climate period (pre-industrial up to now) between

the two models, whereas the change in risk of future climate scenarios show good agreement.

Fig. 7b shows return periods from the Historical, Current, Natural,1.5 Degree and 2.0 Degree simulations, which we use to analyse how probabilities may change in the future with respect to now. The Current and Historical ensembles are very similar as expected as both are all forcings simulations of differing (but overlapping) lengths. Under 1.5 and 2° C of additional warming, high precipitation within the region is set to increase with risk ratios (compared to Current) derived using the CPC

observational threshold of 1.46 (95% CI 1.27 to 1.69) and 1.74 (95% CI 1.52 to 1.99) respectively. In both cases the ERA-int (GPCC) threshold risk ratio is smaller (larger) than the CPC threshold risk ratio (not shown), but with overlapping uncertainty bounds with CPC. For 2° C of warming these risk ratios show good agreement with the EC-Earth values.

## 4.2 Discharge

In this section we present model validation and results of the discharge simulations, first for the model PCR-GLOBWB, and

then for SWAT, Lisflood and RFM.

The runs with the PCR-GLOBWB model are treated in the same way as the EC-Earth runs. The experiment in which the PCR-GLOBWB model is driven by CPC precipitation and ERA temperature and evapotranspiration shows a strong trend in discharge, which was not seen in the discharge observations. The GEV fit parameters encompass the best estimate from





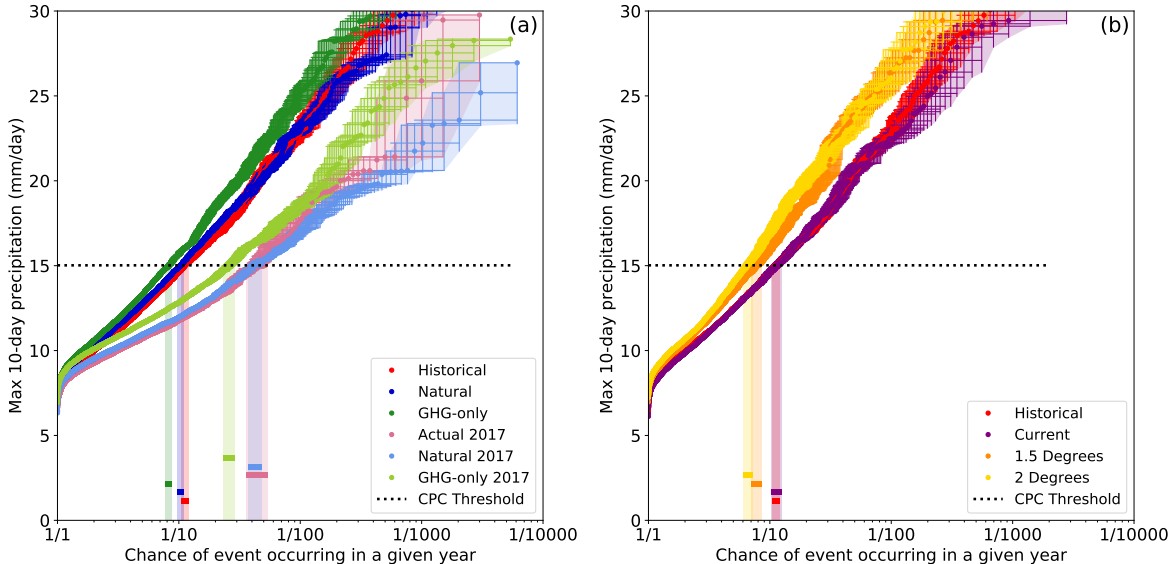

**Figure 7.** Return times of the maximum 10-day precipitation from weather@home simulations. (a) shows results from the Historical, Natural, GHG-only and Actual 2017, Natural 2017 and GHG-only 2017 simulations and (b) the Historical, Current, 1.5 and 2 Degree simulations. Black horizontal lines represent the threshold values derived from the CPC observations. Shaded coloured vertical boxes with solid horizontal lines represent the uncertainty in the return period for the CPC threshold.

observations when fitted with a trend. However the large discharge events of 1988 and 1998 are not captured in this run (not shown).

The experiment with ERA input, on the other hand, shows no trend but does clearly show the strong discharge events of 1988 and 1998 (not shown). The best estimate of the GEV fit parameter is outside the error margins of the GEV fit parameters

of observations, however, the error margins do overlap.

These two model runs show that the PCR-GLOBWB model is able to capture historical flood events but the magnitude of these event is dependent on the meteorological input data. Furthermore, we find that the statistical properties are a fair representation of the statistical properties of observed discharge.

We perform an additional validation for the transient PCR-GLOBWB run with EC-Earth 2.3 input over the years corre-

sponding to years with observed discharge. With this input the modelled discharge peaks in August, but is also high in July and September. We thus use the same months JAS as in observations for further analysis. Different from the observed distribution, the shape parameter $\xi$ is positive, showing higher discharge values in the tail. This is not a problem for this analysis, as the return period of about 4 years that we are interested in is not in the tail of the distribution. When comparing the error margins of the ratio $\sigma/\mu$ with observed statistics we note that the model variability is too large compared to the model mean. This is

not the ideal situation and we note in the discussion how this model bias affects the analysis.





**Table 3.** Risk ratios for precipitation and discharge, for models and observations for both present against pre-industrial or 1900 and a 2° C climate against present. 95% confidence intervals are given as well.

| | dataset | RR (present-pre-industrial or 1900) | 95% CI | RR (2° C- present) | 95% CI |
|---|---|---|---|---|---|
| precipitation | CPC | 18.2 | >0.20 | | |
| | EC-Earth2.3 | 3.27 | 2.65 ... 4.24 | 1.81 | 1.75 ... 2.14 |
| | w@h (Historical/Natural) | 0.92 | 0.84 ... 1.02 | 1.74 | 1.52 ... 1.99 |
| | w@h (GHG-only/Natural) | 1.73 | 1.34 ... 2.25 | | |
| | w@h (GHG-only/Historical) | 1.35 | 1.23 ... 1.49 | | |
| | w@h (Actual 2017/Natural 2017) | 0.97 | 0.76 ... 1.23 | | |
| | w@h (GHG-only 2017/Natural 2017) | 1.65 | 1.38 ...1.96 | | |
| | w@h (GHG-only/Actual 2017) | 1.69 | 1.35 ... 2.13 | | |
| discharge | observations | 1.43 | 0.05 ... 42.5 | | |
| | PCR-GLOBWB (EC-Earth) | 2.34 | 1.74 ... 2.37 | 1.34 | 1.23 ... 1.41 |
| | SWAT - EC-Earth (transient) | 1.49 | 1.30 ... 1.57 | 1.56 | 1.45 ... 1.7 |
| | SWAT - w@h (Actual 2017/Natural 2017) | 0.88 | 0.72 ... 1.09 | | |
| | Lisflood - w@h (Actual 2017/Natural 2017) | 1.35 | 1.20 ... 1.51 | | |
| | Lisflood - w@h (GHG-only 2017/Actual 2017) | 1.29 | 1.10 ... 1.45 | | |
| | Lisflood - w@h (GHG-only 2017/Natural 2017) | 1.74 | 1.52 ... 2.01 | | |
| | RFM - w@h (Actual 2017/Natural 2017) | 1.13 | 1.11 ... 1.14 | | |
| | RFM - w@h (GHG-only 2017/Actual 2017) | 1.53 | 1.50 ... 1.56 | | |
| | RFM - w@h (GHG-only 2017 /Natural 2017) | 1.73 | 1.71 ... 1.74 | | |

Using the transient model runs, the risk ratio of discharge is calculated in the same way as for observations, using all data between 1880-2017. The risk ratio between 2017 and pre-industrial is 2.3 (95% CI 1.7 ... 2.4), see Fig. 8. For the future we calculate return periods from the present and future distributions separately, following the same statistical method as for precipitation in the EC-Earth 2.3 present and future experiments. The risk ratio between a 2° C climate and the present follows from this, with a value of 1.3 (95% CI 1.2 ... 1.4). We thus conclude that in the PCR-GLOBWB model driven by EC-Earth output there is a positive trend in discharge events like the one in August 2017 in both the historical period (pre-industrial to 2017) and the future period (from current conditions to a +2° C world).

The SWAT model calibrated with EC-EARTH meteorological data tends to underestimate flows in almost all months of the year, see Fig. S3 in the Supplement. The SWAT model calibrated with weather@home meteorological data, on the other hand, tends to underestimate flows in the monsoon months while overestimating flows in the remaining months. Therefore in both cases, flows in our months of interest (JAS) are always slightly underestimated, but the magnitudes of error appear limited enough for the models to be useful in conducting attribution studies. When comparing the error margins of the ratio $\sigma/\mu$ with





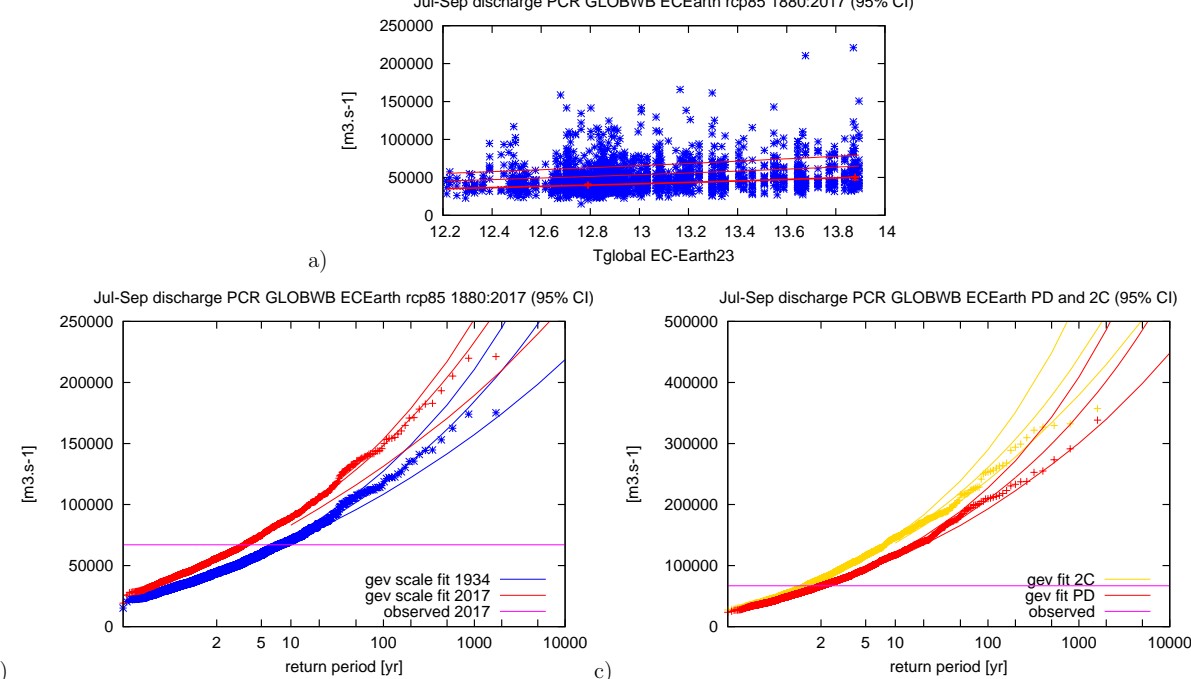

**Figure 8.** Analysis of the highest discharge at Bahadurabad in July–September in the PCR-GLOBWB model over the years 1920-2017. a) the location parameter $\mu$ (thick line), $\mu+\sigma$ and $\mu+2\sigma$ (thin lines) of the GEV fit of the discharge data. The vertical bars indicate the 95% confidence interval on the location parameter $\mu$ at the two reference years 2017 and 1934. b) the GEV fit of the discharge data in 2017 (red lines) and 1934 (blue lines), assuming a trend. The observations are drawn twice, scaled up with the trend (smoothed global mean model temperature) to 2017 and scaled down to 1934. c) GEV fits for the present day (PD, red) and $+2°$ C world (2C, yellow) simulations. The purple horizontal lines in b) and c) show the threshold value for which the risk ratio is calculated.

observed statistics we note that the model variability is too small compared to the model mean, opposite to what was found for the PCR-GLOBWB model. The shape parameter $\xi$ is of the same order as the one in the observed discharge dataset.

The risk ratios are calculated from return period plots for both the EC-Earth-runs (see Fig. 9) and the weather@home-runs (see Fig. 10). Using the SWAT model runs with EC-Earth transient data, we see that the discharge shows some decadal

5  variability. The trend in the data therefore depends more strongly on the years used. For consistency we use the same years as in the analyses of EC-Earth and PCR-GLOBWB data (1880-2017), and we note that the error margins do not capture this variability and are underestimated. The risk ratio of discharge between 2017 and pre-industrial is found to be 1.5 (95% CI 1.3 ... 1.6). The risk ratio between a $2°$ C climate and the current climate is 1.56 (95% CI 1.45 ... 1.70). Using the SWAT model runs with weather@home Actual 2017 and Natural 2017 data, the risk ratio between the Actual 2017 and Natural 2017 scenario is

10  0.88 (95% CI 0.72 ... 1.09).





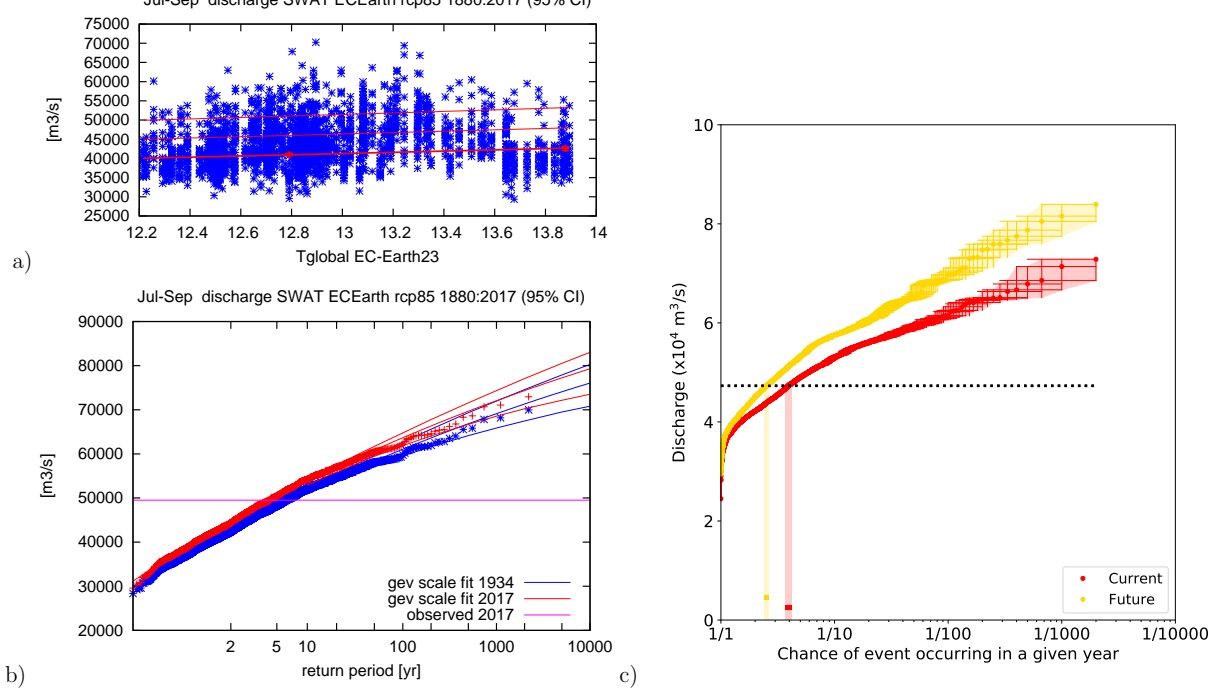

**Figure 9.** Analysis of the highest discharge at Bahadurabad in July–September in the SWAT flows for EC-Earth. a) the location parameter $\mu$ (thick line), $\mu + \sigma$ and $\mu + 2\sigma$ (thin lines) of the GEV fit of the discharge data. The vertical bars indicate the 95% confidence interval on the location parameter $\mu$ at the two reference years 2017 and 1934. b) the GEV fit of the discharge data in 2017 (red lines) and 1934 (blue lines), assuming a trend. The observations are drawn twice, scaled up with the trend (smoothed global mean model temperature) to 2017 and scaled down to 1934. c) Current and future simulations. The purple horizontal line in b) and dotted line in c) show the threshold value for which the risk ratio is calculated.

Calibration and validation graphs for Lisflood and RFM are shown in the Supplement. They show that both Lisflood and RFM models are able to simulate the seasonality of rise in spring and summer flows correctly. Both models underestimate the river discharge in summer with an underestimation in the simulated discharge by Lisflood.

The return period and risk ratio for the Lisflood model and RFM estimated from the weather@home Actual 2017 and Natural 2017 datasets are shown in Fig. 11, as well as the results for the GHG-only 2017 runs.

The Lisflood model shows that a discharge value with a return period of 4 years in the actual scenario would increase to 5.4 years in the natural climate scenario (risk ratio of 1.35 (95% CI 1.20... 1.51)), while it would reduce to 3.1 years in the GHG-only scenario.

The trend is similar in the results simulated by the RFM, however, the discharge value with a return period of 4 years is slightly greater than the value simulated by Lisflood. The return period would increase to 4.5 years under natural climate conditions (risk ratio of 1.13 (95% CI 1.11... 1.14)), while it would reduce to 2.6 years in the GHG-only scenario. Note however





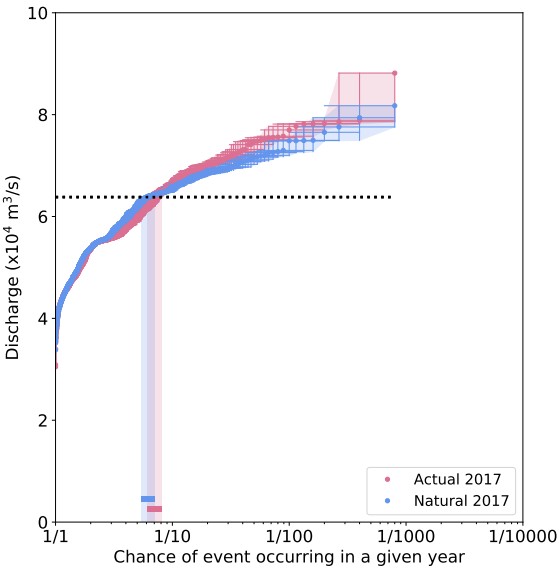

**Figure 10.** Return period plots for SWAT flows with weather@home data for the Actual 2017 and Natural 2017 ensembles

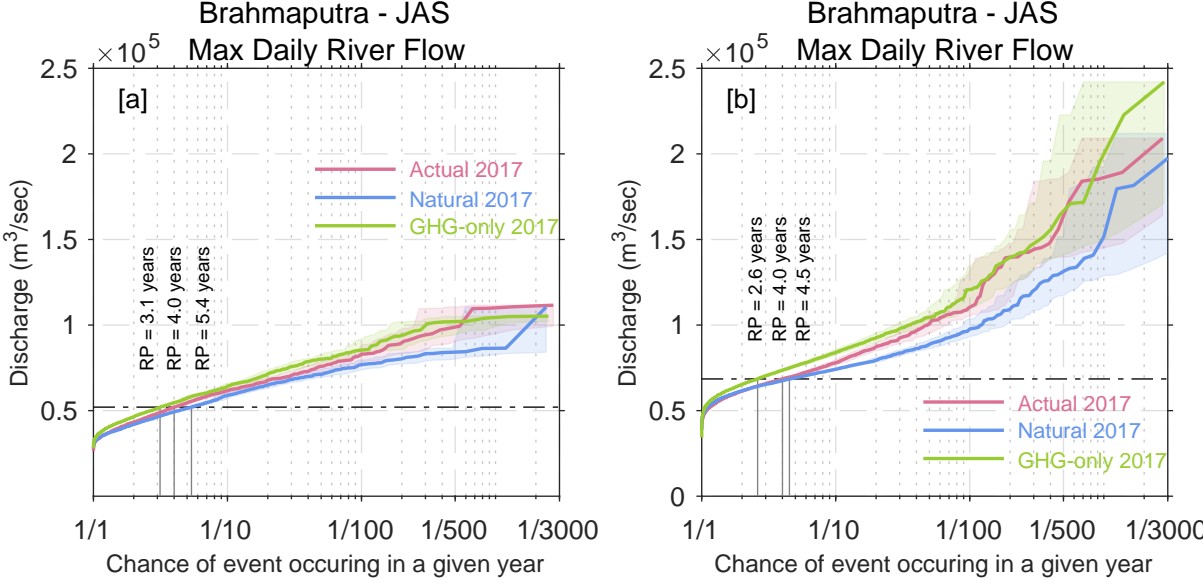

**Figure 11.** River flow return periods simulated by (a) Lisflood and (b) RFM model using the Actual 2017, Natural 2017 and GHG-only 2017 scenarios.





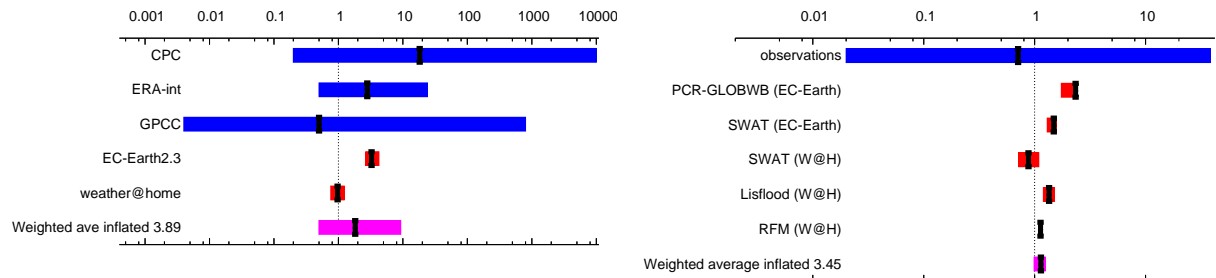

**Figure 12.** Synthesis of the precipitation (left) and discharge (right) results. Dark blue is observations, red is climate model ensembles and the weighted average is shown in purple. The ranges of the models are not compatible with each other, pointing to model uncertainty playing a role over the natural variability. The weighted average has been inflated by a factor 3.89 and 3.45 for precipitation and discharge respectively to account for the model spread.

that from Fig. 11b we see that the risk ratio between the different scenarios for RFM becomes larger for larger return periods (e.g., 10 years) than studied in this analysis.

The shorter return period in the GHG-only 2017 scenario shows that if sulphate aerosols are removed from the atmosphere (which results in increased precipitation), flooding becomes more frequent. This implies that floods can become more frequent
in the region if the air pollution levels are reduced in the future.

The risk ratios for the observed threshold from both Lisflood and RFM of 1.35 (95% CI 1.20... 1.51) and 1.13 (95% CI 1.11...1.14) respectively are in good agreement even though the simulated river flows by the models are different. The mitigation effect due to the aerosols is also comparable between these two different hydrological models.

## 5    Synthesis

In observations the uncertainties in return periods and risk ratios are quite large. This is mainly due to the shorter lengths of the time series, and natural variability dominates. In the models, the signal to noise ratio is much larger, resulting in smaller uncertainties in the risk ratios. Here, the model spread dominates the signal. As both natural variability and model spread play a role, we use a weighted average with inflated uncertainty range. We do not synthesize the risk ratios for the future, as we only have two model estimates per variable.

In the synthesis we use all available observational datasets that are analysed in this paper and one experiment per model. For weather@home and all hydrological models that use input from weather@home experiments we use the risk ratios calculated from the Actual 2017 and Natural 2017 experiments. This gives a fair opportunity to compare the synthesis of precipitation with the synthesis of discharge.

The synthesis results are shown in Fig. 12. The synthesis of the precipitation analysis results in a risk ratio between 2017
and pre-industrial of 1.8 (95% CI 0.5 ... 9.3). Although the best estimate is above one, the trend is not significant due to the relatively large error margins. The synthesis of the discharge analysis results in a risk ratio between 2017 and pre-industrial of





1.1 (95% CI 1.0 ... 1.3). So for discharge the best estimate is only slightly higher than one, and due to the smaller error margins in the average, this trend is just significant under the assumptions made in this analysis.

## 6 Discussion

In any event-attribution study, tasks to be carried out include;

(i)  determining what happened using available observations and defining the event to be studied,

    (ii)  determining how rare the event is in current and pre-industrial conditions,

   (iii)  using models to attribute any changes in likelihood of similar classes of events.

Here we discuss some of the issues encountered in these steps and the interpretation of our results in the light of uncertainties.

First of all, determining the amount of precipitation falling into the Brahmaputra basin from observations (and thus the ap-
propriate precipitation threshold to define this event) is not trivial. As is common in regions with strong topographic gradients, estimating area-averaged rainfall based on observed rainfall is challenging, as rainfall differences between neighbouring locations can be very large in reality, and the orography, which is only partly resolved by a sparse observational network (or model grid), drives these differences. A large part of the Brahmaputra basin has an elevation of over 2000 m hence unsurprisingly different precipitation datasets show very different spatial and temporal characteristics. They are all likely to underestimate the
precipitation at higher elevation, where few weather stations record data (Immerzeel et al., 2015).

For this analysis we used the CPC dataset to provide a single estimate of the event magnitude (i.e. determine what happened) and to define the return period (i.e. determine the rarity of the event) for use in the other data sets and models. Applying this return period, we used three observational data sets to convey the uncertainty related to observations in the resulting risk ratios. However, for the GPCC dataset, the very limited temporal length of the record leads to an uncertainty estimate that is too high
for meaningful inference on the change in risk to be made. The longer records do show an increase in the chance of extreme rainfall but again uncertainties affect a clear signal detection. The intended future availability of high-resolution reanalyses such as ERA5 that will cover the years 1950 onwards at 30 km resolution will potentially improve trend analyses in high-mountain regions in Asia.

From the hydrological perspective, we defined the event as the maximum daily discharge at Bahadurabad in July–September.
In contrast to precipitation data, there is only one official discharge observation series, which does not allow for intercomparison. The determination of flood risk, however, appears sensitive to the hydrological variable under study. To obtain an impression of this sensitivity, we checked how discharge compares to water level, as a second measure for the likelihood of flooding. The return period of the measured 2017 discharge peak is indeed lower than the return period of the measured 2017 water level peak. Several factors could have influenced this. First of all, the Brahmaputra is a highly braided river and during severe
flood events water enters the floodplain, making it more difficult to accurately relate water level measurements to discharges estimates. Therefore though the water level records are very accurate, the discharge records are unlikely to be of the same





accuracy. Based on the observation of the massive spatial extent of the 2017 floods both in Bangladesh and India, we opine that the observed discharges are likely higher than those recorded.

This opinion is supported by the change in correlation between discharge and water level. The correlation between water level and discharge is 0.88 over the whole time series. However, after 2011 this correlation changes to almost one, with a

tendency toward discharges values that are lower for similar water levels than before this change. This change could be due to recalibration of the relationship between discharge and water level. We therefore expect that the true return period is between the return period calculated from discharge given above and the return period calculated from water level. As we do not know the exact influence of the change in measurement method of discharge on the discharge values, we cannot give more precise values.

However it should be noted that ongoing morphological changes can introduce additional variability along the river. For instance, higher water levels with lower discharges may be caused by silting and narrowing of the river. McLean and O'Connor (2013) already showed that, for the years 2006-2011, the relation between discharge and water level changes over time; in 2011 similar discharge values lead to higher water levels. This leads to a non-climatic trend in the water level observations.

Climate models, while far from perfect in their representation of reality, are essential to interpret the results from observations

and thereby attribute any observed changes in event frequency to anthropogenic climate change or other factors. Taken at face value, the two climate model simulations of 10-day precipitation maxima in the Brahmaputra basin provide somewhat contradictory results. However, for the weather@home simulations when comparing the natural simulations with GHG-only runs instead of historical simulations, the change in extreme precipitation is significantly positive as well and therefore better comparable in magnitude to the increase in the two longer observational datasets and EC-Earth simulations. Comparing the

GHG-only runs to the historical simulations gives an indication of the impact of aerosol within the weather@home model which might be slightly overestimated given that black carbon is not included in the models aerosol treatment. Nonetheless, HadRM3P clearly indicates that the increased risk in extreme rainfall due to GHG induced warming has been effectively counterbalanced by aerosol emissions. The EC-Earth model is interpreted to have less aerosol effects and hence to show more of the greenhouse gas-driven increase. Both results are in agreement with the observations due to the large uncertainties in the

limited-length observational records.

The counterbalance between the greenhouse gas and aerosol effects may also be important for clean air policy decisions: as the air is cleaned the already committed increase in extreme precipitation due to greenhouse gases will be revealed. These results also suggest that the overall signal from long term climate change, i.e., mainly greenhouse gas forcing, in the datasets where we cannot separate out the impact of aerosol forcing might be underestimated. The best estimate of the change in risk

in extreme rainfall as observed in the Brahmaputra basin in 2017 is therefore likely a rather conservative estimate and hence of limited use to inform decision making. In fact, simulations of the near future of both models show a clear increase in the risk of high-precipitation events that lead to flooding on the Brahmapura.

In extending our multi-method attribution approach to include hydrological modeling, we consequently introduced more degrees of freedom in possible combinations of inputs and models to construct the hydrological response. Time and compu-

tational restraints put a limit on the number of combinations that could be explored. We conducted experiments using (i) the




same hydrological model (PCR-GLOBWB) run at different resolutions with different input observational/modelled meteorological input data, (ii) the same input climate model (weather@home) with different hydrological models and (iii) the same hydrological model (SWAT) with two different input climate models. Changing the resolution of the PCR-GLOBWB runs with CPC and ERA-int input compared to runs with EC-Earth 2.3 input, impacts the dynamics in the hydrological model. In general

coarser resolution simulations respond faster due to the decrease in storage and the shorter connectivity between gridcells. High resolution models are better able to capture the subsurface and riverine water storage due to their increased heterogeneity (Sutanudjaja et al., 2017). It is therefore more difficult to simulate extreme hydrological events in coarser models (Samaniego et al., 2018). It was beyond the scope of this paper to analyse the differences in detail, however, we use the differences to show the range of possible output within one hydrological model. None of the models or observational datasets are perfect. For in-

stance, in the PCR-GLOBWB model the variability is too high compared to the mean, while RFM and Lisflood underestimate the magnitude considerably. This is not the ideal situation however, there is no reason to believe that the order of magnitude of the risk ratios between the current and past climate or between the future and current climate will depend on this very strongly. This is corroborated by the fact that the risk ratios are comparable despite the very different biases.

Despite these strong differences in variables, resolution, simulated processes as well as input data, the simulated changes

in the likelihood of the observed event occurring because of anthropogenic climate change are very comparable. Even when the hydrological models are driven by precipitation from the weather@home simulations the simulated discharge shows a significant increase in likelihood apart from SWAT where the change is not significant.

## 7 Conclusions

In August 2017, following heavy rains, Bangladesh faced one of their worst river flooding events in recent history, with record

high water levels leading to inundation of river basin areas in the northern parts of the country, impacting millions of people who are highly exposed and vulnerable to unusual flooding.

This paper presents an attribution of this precipitation-induced flooding event and, for the first time, extends the multi-method approach of extreme event attribution from a purely meteorological perspective to the more impact-relevant hydrological perspective, by employing an ensemble of hydrological models. Firstly, experiments were conducted with three observational

data sets and two climate models to estimate changes in extreme precipitation event frequency, in the 10-day Brahmaputra basin average, that have occurred since pre-industrial times. In addition, climate projection experiments were used to indicate if the trends found up to now are likely to continue or become more extreme in the future. The precipitation series were then used in turn as meteorological input for four different hydrological models to estimate the corresponding changes in river discharge. In doing so, a range of possible answers to the attribution question were produced, allowing for comparison between approaches

and for the robustness of the attribution results to be assessed.

Specifically, our aims were to (i) determine if precipitation can be used as a measure of the extremity of flooding in the large Brahmaputra basin, or if it is necessary to instead use a hydrological measure such as discharge, for the purpose of attributing





the flood of August 2017 in Bangladesh, (ii) conclude on the attribution of this event, expressed as the change in likelihood, of similar or more extreme events, that has occurred since pre-industrial times and which is projected to occur in the future.

From the precipitation perspective, we find that two out of three of the observed series show an increased probability for extreme precipitation like observed in August 2017, but in all three observational data sets the trends are not significant due to

the short records. One climate model shows a significant positive influence of anthropogenic climate change, whereas the other simulates a cancellation between the increase due to greenhouse gases and a decrease due to sulphate aerosols. The change in risk of high precipitation that has occurred since pre-industrial times is therefore uncertain. However, both climate models agree that the risk will increase significantly in the future, by more than 1.7 with 2° C of global heating since pre-industrial times.

Considering discharge rather than precipitation, which corresponds more closely with the hydrological impacts, shows only a slightly different result, in that the increase in risk since pre-industrial times to current day of high discharge synthesized from both observations and models is just significant, whilst the risk of high precipitation is not. The attribution of the change in discharge is therefore somewhat less uncertain than for precipitation, but the 95% CI still encompasses no change in risk. For the future, these models project a slightly smaller increase in probability of high discharge than of high 10-day precipitation,

being about a factor 1.5 more likely in 2° C warmer world.

For large basins in orographically diverse regions with complex hydrology, such as the Brahmaputra, we hypothesized that rainfall, river flow and inundation would not be linearly connected and that precipitation would not be an adequate measure of flood intensity. The initial hydrological conditions play an important role in combination with the occurrence of high intensity precipitation events. We therefore anticipated that small changes in the risk of precipitation would lead to disproportionate

changes in flood risk, evidenced in differences in the risk ratios of the event calculated from the two perspectives.

Our synthesis, however, produces a best estimate for the past climate that is greater than one and of similar order of magnitude (between 1 and 2) for both methods, and a lower bound on the uncertainty range that is less than or about equal to one, leading to the conclusion that we cannot confidently confirm a significant anthropogenic influence in changes up till now. Projected changes between current conditions and for a 2° C warmer-than-preindustrial world were also a similar order of magnitude

(between 1 and 2) for 10-day precipitation and discharge, with significant changes found. Thus, in this particular case, studying precipitation alone would have led to the same qualitative conclusion.

Inspecting the individual model outcomes shows that in the study of this particular event, there is an impact of the choice of circulation model used as input for the hydrological model on the amplitude of discharge RR's. Where the EC-Earth model was used, we find a larger positive change in precipitation compared to discharge, but where the weather@home model was

used, we find a similar or smaller positive change in precipitation compared to discharge. This highlights the importance of using multiple models in attribution studies, particularly where the climate change signal is not strong.

The use of multiple methods in the attribution of extreme events is the only way to estimate confidence in attribution results and hence reliability. As hydrological models are used to simulate impact-relevant variables (such as flood depth) and are in fact used much more for decision making, it is essential to extend the attribution approach in general to include hydrological

models when possible, for analysis of precipitation-induced flood events. Hydrological models offer further insight into the



partitioning of precipitation reaching the ground, and thus come closer to the drivers of the impacts observed on people and livelihoods. Climate models on the other hand allow us to disentangle the potential effects of different atmospheric drivers.

This highlights that only a combination of doing a multi-method attribution analysis of the meteorological drivers with a multi-model approach in hydrological modelling allows for a robust estimate of changing flood hazards under climate change.

Therefore we recommend the use of a hydrological variable, such as discharge, for estimating changing flood risk in large basins such as the Brahmaputra, although based on this study, investigating changes in precipitation is also useful, either as an alternative when hydrological models are not available, or as an additional measure to confirm qualitative conclusions.

*Data availability.*   Almost all data are available under https://climexp.knmi.nl/selectfield_att.cgi

*Acknowledgements.*   SS, HJ, KM, DW, FO and SI were funded as part of the EPSRC GCRF Institutional Sponsorship REBuILD project.

KvdW was funded as part of the HiWAVES3 project. NW acknowledges the funding from NWO 016.Veni.181.049. This work was partially supported by the EUPHEME project, which is part of ERA4CS, an ERA-NET initiated by JPI Climate and co-funded by the European Union (Grant 690462). We would like to thank the Met Office Hadley Centre PRECIS team for their technical and scientific support for the development and application of weather@Home. We are grateful to Dr. Simon Dadson and Homero Paltan Lopez for sharing the RFM model code and for their help in setting it up for the study area. Finally, we would like to thank all of the volunteers who have donated their

computing time to climateprediction.net and weather@home.



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
