# Peer review of "Attributing the 2017 Bangladesh floods from meteorological and hydrological perspectives"

_Hydrology and Earth System Sciences, 2018_

## Referee Comment (RC1) · Anonymous Referee #1 · 28 Aug 2018

Dear Editor, The submitted manuscript entitled "Attributing the 2017 Bangladesh floods from meteorological and hydrological perspectives" is a well written and structured paper. They have analyzed 10-day precipitation index for extreme events in August as well as river discharge over Brahmaputra basin. I have a few comments for improving the paper:

It is unclear why 10-day average precipitation is considered where the 1-day or 5-day maximum precipitation are well known as flood index.

Please explain the role of temperature in precipitation change. Based on the ground observations, can you explore a relationship between them over the study area?

I have a concern about the validity of scaling the GEV parameters (location and scale)

[Figure]

similar to Clausius-Clapeyron (CC) relationship in the context of urban climate. The observed global mean surface temperature (GMST) is a feature confined to the boundary layer, whereas, precipitation is formed in clouds that develop in the free atmosphere up to a height of several kilometres, so it is unlikely that the surface temperature has some effects on precipitation in terms of the CC relationship. I would therefore recommend making the physical meaning of this scaling clearer.

Moreover, it is not clear the CC relationship exhibited by 10-daily extremes in your study area linked with convective nature of precipitation.

Add some details into the Statistical methods for trend detection. Time series of parameters are may be autocorrelated (temporal dependency over times scales of several years). I am wondering whether the authors took these autocorrelations into account or not.

---

## Referee Comment (RC2) · Anonymous Referee #2 · 19 Oct 2018

Cause of the severe floods and their occurrence tendency in relation to climate change are very important topic in Asia, and quantitative assessments with in-situ data analysis with model verifications are expected. This study focused in the severe flood in 2017, and counterbalances of precipitation and discharge for the long-term return periods of floods are discussed based on observation data and multiple model output. I admire the author's challenges with many works, however, I could not capture (understand) the fundamental objectives and clear results from the paper. As presented in the title, "attribution" and "perspective" mislead the readers to know the target of the paper. I would like to suggest fundamental revisions of the paper.

Major comments 1) Ambiguous objectives and results Major objectives may be indicated in paragraphs at P4L25 (or P26L14), such as not only the precipitation vari-

ability but ability of discharge needs to be considered for estimating return period of floods. However, there are no explanations about the physical mechanism (perspective?) of extreme precipitation/discharge variability to cause flood events in Bangladesh based on hydro-climatological point of view with references. Regarding to the long-term changes of the ability of discharge in such a large scale watershed, they would be strongly related to changing of river sediments, micro-topography affected by previous floods, artificial settlements such as bridges or bank, or expansion of residences due to population increase. I could not understand how the long-term trends or probability for return periods of precipitation/discharge could link to an extreme event without assessments to cause flood in 2017 as a case study. Many considerations are discussed, however, clear results are not show, such as "trends are not significant (P1, L7; P26L19)", "values are less uncertain, (P1L12; L26L28)", "cancelation between A&B (P26L21)", etc.

2) Descriptions of chapters are like reports, not as in article. This study prepared several kinds of observation or model based data, and analyzed long-term probability. Do you want to compare something or ensemble to produce better predictions? Forecast map of Aug. 2017 was already shown in Fig. 1, but what is the problem for this prediction? In the Section 3, observation of water level was additionally analyzed, but why the analysis of discharge is not enough? Usually, discharge is calculated based on water level, and there is not discussion of water level in the section of "Model analysis". In many parts, the authors described all the matter of what they did as reports, but reader can not capture reasons and corresponding results based on logical explanation.

3) Many careless parts for reader. Explanations are insufficient and can not understand the explanations in many parts. I would like to ask brush up the paper again, such as; > Where is a green circle, Brhamaputra, too small words in Fig.1 > What is the "attribution methods" at P4L30? > What the scale of "past, present future" stand for ? (P4L33). > You use CPC data "for what?", P5L17. > What is the "same analysis" at P6L8? > Confusion of the study order exists, such as in 2.2.1, such that Use 3

experiments Transient experiment Two time slice experiments Large ensembles are created First set of experiments,,, A second climatology,, A third ensample,,, The second set of experiments A third set of experiments,,, > "several river discharge simulations" at P8L28, corresponds to five different model experiments or several different setting in the same model? Or, simulations at several rivers? > Several figures are shown without explanation, such as Fig.3b, Fig.5a, etc. > "Large uncertainty in the accuracy of data " at L6P13, how you detect them and why you used them? > Etc..

---

## Author Response (AR3)

Final authors response to **'Attributing the 2017 Bangladesh floods from meteorological and hydrological perspectives'** by Sjoukje Philip et al., 2019.

We would like to thank the editor and all reviewers for their careful reading.

This document contains the replies to all reviewers, including the interactive discussion, and a marked-up version including all changes between the first version of the discussion paper to the current final version.

Dear Editor,

We will answer your questions below  and we attached the replies to the reviewers that are also online in the open discussion to this explanation. Finally, this document contains a marked-up manuscript version.

*Referee 2 is quite critical in the review so I would prefer that you give more thoughts on the substantial issues and try to see if they can be addressed. The current explanation is fine for the open discussion part.*
In addition to the reply to reviewer 2 we made some extra changes to more clearly state the purpose of the paper. For instance in the abstract we add that

1. we do an attribution of this precipitation-induced flooding '**to anthropogenic climate change**'
2. we estimate changes in extreme 10-day precipitation event frequency over the Brahmaputra basin '**up to the present**' to avoid confusion with future projections.

And in the introduction we explain the word 'perspectives' by adding: Thus we explore the drought in two different ways - first from a meteorological perspective (using precipitation data) and then from a  hydrological perspective (using discharge data).

Furthermore, we put more emphasis on the fact that we for the first time do a multi-model multi-method attribution study in this paper. We emphasized this by adding in the introduction:
Schaller et al. (2016) already studied a flooding case in an attribution study using one hydrological model. In this paper we for the first time do an attribution study using observational precipitation and discharge data and a combination of GCMs and hydrological models.

Finally we moved most of the details on the models to the supplement.

*If you prefer you may keep the title as is.*

Thank you, based on your opinion and the additions we made as explained above we indeed decide to keep the title as is.
*Personally I appreciate your work very much but also realize that is rather a daunting task to sort out all the related issues both in the atmospheric and the surface part that caused the floods. An initial thought was if you could construct some joint probability distributions for the events and evaluate the likelihood of future events under certain plausible conditions. Some certain types of IDF (intensity-duration-frequency) may be helpful to link the probabilities of the precipitation and the floods.*

As you remark, we already had to deal with a daunting array of variables. Our decision to use 10-day averaged precipitation for the Brahmaputra basin was based on Fig. 5 from a paper by Webster et. al. (2010), which we already refer to in our the paper. We attached this figure to our reply and now also added the reference to this figure in our paper as well. Taking longer time averages (e.g. 15 days) would not only include precipitation from far away in the basin from 15 days ago, but also from 15 days ago near the flooded area. Similarly, taking shorter averages (e.g. 5 days) would exclude the rainfall from far away in the basin that travels more than 5 days to the flooded area. The choice to use a 10-day average is a compromise that seems to fit the purpose of our analysis quite well. Limiting the analysis to the 10-day precipitation scale, which corresponds well with the observed discharge (other time scales

gave lower correspondences) and from the hydrological model, was one way to limit the study to a manageable size. Computing full IDF diagrams would introduce even more variables without shedding much additional light on the central questions: is there a discernible climate change signal in the Bangladesh floods and how much value do hydrological models add.

We hope for a positive decision.

kind regards,
Sjoukje Philip

[Figure]
*Dear Editor, The submitted manuscript entitled "Attributing the 2017 Bangladesh floods from meteorological and hydrological perspectives" is a well written and structured paper. They have analyzed 10-day precipitation index for extreme events in August as well as river discharge over Brahmaputra basin. I have a few comments for improving the paper:*

*It is unclear why 10-day average precipitation is considered where the 1-day or 5-day maximum precipitation are well known as flood index.*

The Brahmaputra basin feeds the rivers in Bangladesh. The basin is so large that using only 1-day or 5-day precipitation would not take the precipitation into account that falls further upstream in the basin. We take a 10-day average to represent the area that collected water that arrives at Bahadurabad and contributed to the flooding. Using 1-day or 5-days over the whole basin would exclude the additional water from region of the Brahmaputra basin in the Northeast of India that reaches Bahadurabad at the same time as precipitation that falls later in the region closer to Bahadurabad.

We know that averaging over such a large basin and time scale is not the most ideal situation. This is why we compared the results to the analysis results from discharge.

*Please explain the role of temperature in precipitation change. Based on the ground observations, can you explore a relationship between them over the study area?*
*I have a concern about the validity of scaling the GEV parameters (location and scale) similar to Clausius-Clapeyron (CC) relationship in the context of urban climate. The observed global mean surface temperature (GMST) is a feature confined to the boundary layer, whereas, precipitation is formed in clouds that develop in the free atmosphere up to a height of several kilometres, so it is unlikely that the surface temperature has some effects on precipitation in terms of the CC relationship. I would therefore recommend making the physical meaning of this scaling clearer.*

The scaling is taken to be an exponential function of the smoothed global mean temperature, This exponential dependence can clearly be seen in the scaling of daily precipitation extremes with local daily temperature in regions with enough moisture availability (Allen and Ingram 2002; Lenderink and van Meijgaard 2008). It is also expected on theoretical grounds through the first-order dependence of the maximum moisture content on temperature in the Clausius-Clapeyron relations of about 7%/K, which gives rise to an exponential form. Note that we fit the strength of the connection, which is often different from CC scaling. As it is not clear what the relevant local temperature is, but local temperature usually scales linearly with the global mean temperature, we chose the latter.

We will add this as a paragraph in the manuscript.

*Moreover, it is not clear the CC relationship exhibited by 10-daily extremes in your study area linked with convective nature of precipitation.*

As we state above, we fit the data using a GEV with scaling to GMST. Comparing the observations to the fit line, we see no evidence that our assumptions are incorrect.

*Add some details into the Statistical methods for trend detection. Time series of parameters are may be autocorrelated (temporal dependency over times scales of several years). I am wondering whether the authors took these autocorrelations into account or not.*

We checked the autocorrelation and found that there is no autocorrelation of the July-September maximum of 10-day mean precipitation, which is the measure we use in this study. (And for single days the autocorrelation becomes negligible within 4 days.) Therefore, we will add in the Statistical methods section for trend detection: We checked that year-on-year autocorrelations of RX10day are negligible, so serial autocorrelations are not a problem in this analysis.

Autocorrelation of annual 10-day mean pr CPC Brahmaputra (iprcp cpc daily mask0 su max1 10v anom) 1979:2017 (eps, pdf, raw data)

[Figure]

Autocorrelation of Jul-Sep precipitation pr CPC Brahmaputra (iprcp cpc daily mask0 su) 1979:2017 (eps, pdf, raw data)

[Figure]

*Interactive comment on Hydrol. Earth Syst. Sci. Discuss., https://doi.org/10.5194/hess-2018- 379, 2018.*

Cause of the severe floods and their occurrence tendency in relation to climate change are very important topic in Asia, and quantitative assessments within-situ data analysis with model verifications are expected. This study focused in the severe flood in 2017, and counterbalances of precipitation and discharge for the long-term return periods of floods are discussed based on observation data and multiple model output. I admire the author's challenges with many works, however, I could not capture (understand) the fundamental objectives and clear results from the paper. As presented in the title, "attribution" and "perspective" mislead the readers to know the target of the paper. I would like to suggest fundamental revisions of the paper.

AC2:
In this paper we perform an attribution analysis starting with precipitation as well as discharge. We think the reviewer interpreted the word 'perspectives' in a different way than we meant it to be. We changed this to 'standpoints', assuming that the title is now more clearly related to the content of the paper.

--
RC2:
Major comments 1) Ambiguous objectives and results Major objectives may be indicated in paragraphs at P4L25 (or P26L14), such as not only the precipitation variability but ability of discharge needs to be considered for estimating return period of floods. However, there are no explanations about the physical mechanism (perspective?) of extreme precipitation/discharge variability to cause flood events in Bangladesh based on hydro-climatological point of view with references. Regarding to the long term changes of the ability of discharge in such a large scale watershed, they would be strongly related to changing of river sediments, micro-topography affected by previous floods, artificial settlements such as bridges or bank, or expansion of residences due to population increase. I could not understand how the long-term trends or probability for return periods of precipitation/discharge could link to an extreme event without assessments to cause flood in 2017 as a case study. Many considerations are discussed, however, clear results are not show, such as "trends are not significant (P1, L7;P26L19)","values are less uncertain,(P1L12;L26L28)","cancelation between A&B (P26L21)", etc.

AC2:
As stated above, the title may be misinterpreted, and we changed it now.

We understand the concern of the reviewer about river sediment etc. and the inability of the climate models to represent this. In the discussion we do mention that water level is not one-to-one related to either precipitation or discharge. Besides, ongoing morphological changes will influence the flooding. We note this and emphasize that an attribution of the flood in terms of flooded area or affected people is not (yet) possible, however, the use of multiple methods and multiple variables in the attribution of extreme events allows for a robust estimate of changing flood hazards under climate change.

In the current work we differentiate between climate change influencing floodings, which we can study with the models used in this paper, and other factors like morphological changes that we cannot attribute to climate change. An attribution study including these additional changes is not (yet) possible.

In this paper we performed an attribution analysis for the flooding in Bangladesh in 2017 using methods currently available. The reviewer is concerned about the (lack of) clarity in our conclusions  There are several possible answers to the attribution question in general, all of which give useful information. These possible answers include i) the event was made more likely due to anthropogenic climate change, ii) the event was made less likely due to anthropogenic climate change, iii) anthropogenic climate change did not alter the frequency of occurrence of the event and iv) with our current understanding and tools we cannot assess whether and how the event was influenced by anthropogenic climate change. Our conclusions, including information about significance and uncertainties, do therefore give a useful answer to the attribution question. Besides, we compare two ways to look at the attribution question on the flooding event; starting with precipitation and starting with discharge.

--
RC2:
2) Descriptions of chapters are like reports, not as in article. This study prepared several kinds of observation or model based data, and analyzed long-term probability. Do you want to compare something or ensemble to produce better predictions? Forecast map of Aug. 2017 was already shown in Fig. 1, but what is the problem for this prediction? In the Section 3,observation of water level was additionally analyzed, but why the analysis of discharge is not enough? Usually, discharge is calculated based on water level, and there is not discussion of water level in the section of "Model analysis". In many parts, the authors described all the matter of what they did as reports, but reader can not capture reasons and corresponding results based on logical explanation.

AC2:
We are not trying improve the forecast of the 2017 event or increase predictability.  Instead we are looking to quantify (1) how the risk of this event occurring has been altered by climate change (by comparing to pre-industrial) and (2) how the risk of this sort of event is likely to change in future (by comparing to 1.5 and 2 degree future ensembles).

--
RC2:
3)Many careless parts for reader. Explanations are insufficient and cannot understand the explanations in many parts. I would like to ask brush up the paper again, such as;
> Where is a green circle, Brhamaputra, too small words in Fig.1

AC2:
As stated clearly in the figure caption both of these panels are reproductions of figures produced by a 3rd party and as such we are unable to edit these.  The links to the original documents are provided where the reader can enlarge the figure reproduced. The green circle is clearly visible along the Brahmaputra river at the NW of the map shown. We added that the green circle is in the northwest of the map.

--
RC2:
What is the "attribution methods" at P4L30?

AC2:
we will change the sentence to:
To compare the differences between the attribution results for the two variables...

--
RC2:
What the scale of "past, present future" stand for ? (P4L33).

AC2:
We have qualified in the text the scale of past,present and future here.

--
RC2:
You use CPC data "for what?", P5L17.

AC2:
We added a sentence at the beginning of section 2, emphasizing that the use of the data is
explained below:
The explanation of how the datasets are used is detailed in Sect.2.3.

--
RC2:
What is the "same analysis" at P6L8?

RC2:
As stated the same analysis was performed on the model results as was performed on the
observations.  We will add that this analysis will be explained in Section 2.3.

--
RC2:
Confusion of the study order exists, such as in 2.2.1, such that Use 3 experiments Transient
experiment Two time slice experiments Large ensembles are created First set of
experiments„ ,A second climatology„A third ensample„ ,The second set of experiments A
third set of experiments„,

AC2:
We agree that this is confusing and have amended the text for clarity.

--
RC2:
"several river discharge simulations" at P8L28, corresponds to five different model
experiments or several different setting in the same model? Or, simulations at several rivers?

AC2: we explain the simulations in the sentences that follow this statement. We will add 'as
explained hereafter'.

--
RC2:
Several figures are shown without explanation, such as Fig.3b, Fig.5a, etc.

AC2:
Fig3b is described, we apologize for mixing up the order of description in the caption and
corrected that. We checked that Fig.5a already has an explanation. We could not find any
other panels without explanation.

--
RC2:
"Large uncertainty in the accuracy of data " at L6P13, how you detect them and why you used them? > Etc..

AC2:
We discuss this in the discussion section. It is known that observational datasets do have uncertainties. We have to use the best data we have.

In general, we think that we explain the uncertainties from using different observations, different variables, different models, different methods etc. well enough. We emphasize that despite these uncertainties and differences, our analysis, in which we combine a multi-method attribution analysis of the meteorological drivers with a multi-model approach in hydrological modelling, allows for a robust estimate of changing flood hazards under climate change.

Dear Editor,

We would like to thank reviewer #3 for his fresh look. We have taken his suggestions into consideration. Please find our reply to Anonymous Referee #3 below, followed by a marked-up manuscript. Small changes have only been made to the abstract and the introduction, which includes an additional reference. No changes have been made to the supplement and this supplementary document is uploaded again without changes.

Kind regards,
Sjoukje Philip

**Author reply to Anonymous Referee #3**

*To my understanding, this is a revised version for an attribution study. I do carefully read the responses as well as the manuscript, and find previous comments are addressed appropriately. The authors combined climate models and hydrological models to analyze a pluvial flooding event, and found the anthropogenic fingerprint was not detectable both for the 10-day precipitation and basin discharge. The attribution method is very comprehensive, and the uncertainty analysis is rigorously conducted. I would like to recommend for the publication besides addressing a few minor comments below, basically for clarifications.*

We thank the referee for carefully reading our manuscript.

*1. The major focus is to detect anthropogenic climate change signal for the flooding event, but the abstract also mentioned the responses of the extremes to 2-degree warming. The authors found that the anthropogenic climate change is not detectable, but there is a clear response to the 2-degree warming. This makes the conclusions confused. I would suggest focusing on the attribution part, rather than the responses to future warming.*

We understand your concern. However we noticed that there is a lot of interest in statements on the future as well. Besides, the fact that there is no anthropogenic climate change detectable up to now, does not mean that there is no change, only that it is not (yet) detectable. A statement on the future thus adds to the conclusion about the past. We therefore prefer to keep the information on a 2-degree warming in the paper. In the abstract we added a few words to clarify that the 2-deg analysis is an extension to the future:

**Extending the analysis to** the future, all models project an increase in probability of extreme events at 2-degree C global heating since pre-industrial times

*2. Abstract. "One climate model shows a significant positive influence of anthropogenic climate change, whereas the other simulates a cancellation between the increase due to greenhouse gases and a decrease due to sulphate aerosols." I would be better to mention this conclusion is based on large ensemble simulations, not just from two model realizations.*

Thank you for pointing to this. We changed the text into:

One climate model **ensemble** shows a significant positive influence of anthropogenic climate change, whereas the other **large ensemble model** simulates a cancellation between the increase

due to greenhouse gases and a decrease due to sulphate aerosols.

*3. Introduction. "In this paper we for the first time do an attribution study using observational precipitation and discharge data and a combination of GCMs and hydrological models." A few published literatures (e.g., Yuan et al., 2018) could be mentioned when reviewing the combination of GCM and hydrological models for attribution of extreme events.*

We were not aware of this study and are happy to use the reference in this manuscript and future studies on floodings or droughts. We added the reference to the paragraph as suggested:

Yuan et al. (2018) use observations, GCMs and one land surface model with and without land cover change to split the changes in observed streamflow and its extremes into anthropogenic and natural climate change, land cover change and human water withdrawal components.

*4. Is there any oceanic background (e.g., anomaly in the Indian Ocean) for this extreme flooding? In fact, there was also a severe flooding over the middle and lower reaches of the Yangtze River basin in July 2017, which was associated with the decay phase of an El Nino and warming over the Indian Ocean.*

We investigated whether extreme rainfall in the Brahmaputra basin was linearly related to SST over 1979--2017 and did not find any connection, in fact the field significance was lower than expected by chance. The same holds for JJA-averaged precipitation averaged over Bangladesh 1891--2016. Canonical ENSO teleconnection maps also show no linear teleconnection in this season.

We did not add this in the paper, as it may distract from the main message which is on trends related to anthropogenic climate change.

*5. P5L2, "drought" or "flood"?*

Thank you for reading so carefully. This indeed needs to be 'flood', we changed this in the text.

*Reference*

[revised manuscript text omitted]